# The histone H3 lysine 36 demethylase KDM2A/FBXL11 controls Polycomb-mediated gene repression and germ cell development in male mice

KDM2A/FBXL11 is a Jumonji-domain containing lysine demethylase catalyzing the removal of mono- and di-methyl modifications of histone H3 lysine 36 (H3K36me1/2). While *Kdm2a* is required for mouse embryogenesis, its role in adult physiology has been largely unexplored. Using conditional deletion approaches, we demonstrate that *Kdm2a* deficiency leads to testicular atrophy and male infertility. Although spermatogonial stem cells remain unaffected, proliferating and differentiating spermatogonia exhibit delayed cell cycle progression and apoptosis. RNA-sequencing of purified spermatogonia and spermatocytes reveals *Kdm2a*-dependent repression of over 750 genes during spermatogonial differentiation. Chromatin immunoprecipitation followed by sequencing (ChIP-seq) demonstrates increased H3K36me2 levels at CpG-rich gene promoters in *Kdm2a*-deficient spermatogonia. KDM2A is required for Polycomb-mediated repression as shown by increased H3K36me2 and reduced H3K27me3 promoter occupancies and failed gene repression in *Kdm2a* deficient differentiating spermatogonia. Loss of *Kdm2a* in spermatocytes disrupts progression through meiotic prophase, as evidenced by impaired completion of chromosome synapsis and processing of meiotic double-strand breaks (DSBs), by altered chromatin states and by an impairment of X-linked gene repression. Our study thus identifies critical roles for KDM2A in coordinating gene expression programs during spermatogonial differentiation and meiosis, which are essential for male germ cell development.

Spermatogenesis is an orderly cellular differentiation process involving multiple mitotic and meiotic divisions that continuously generate mature haploid gametes required for fertilization of oocytes. Like somatic development, spermatogenesis is tightly controlled by chromatin modifiers, which regulate chromatin states and contribute to the coordinated control of gene expression that is critical for male germ cell development.

Homeostasis of spermatogenesis is maintained through self-renewal of undifferentiated type A spermatogonia in the seminiferous tubules of the testis[1]. The progression of germ cell development within seminiferous tubules of mice can be classified into 12 recurring stages (conventionally noted by Latin numerals I-XII), that are based on the composition of developing germ cells (Fig. 1g)[2,3]. Single type A (As) spermatogonia give rise

✉e-mail: antoine.peters@fmi.ch; thomas.nicholson@novartis.com

to proliferating type A paired (Apr) and type A aligned spermatogonia (Aal-4, Aal-8, Aal-16 or Aal-32), which remain interconnected by cytoplasmic bridges and that can still self-renew. Type As, Apr and Aal spermatogonia can be found in any stage of the seminiferous tubules. At stage VIII, type Aal spermatogonia initiate spermatogonial differentiation, forming subsequent type A1, A2, A3, A4, intermediate (In) and type B spermatogonia[4]. Type

A1-4 spermatogonia are found in stages IX to II, and In spermatogonia are found in stages II-III which differentiate into B spermatogonia in stage IV and V (Fig. 2g). Hence, following six mitotic divisions of differentiating spermatogonia, B spermatogonia form at stage VII preleptotene spermatocytes which after replication enter meiotic prophase at stage VIII to form leptotene spermatocytes. Spermatocytes go through complex processes involving

**Fig. 1 | Testicular atrophy in *Kdm2a* iKO animals. A** Schematic diagram illustrating the targeting and conditional deletion strategies of the *Kdm2a* locus. F1 and R1 represent PCR primers used for genotyping. Locations of FRT and loxP sites are indicated. **B** Tamoxifen dosing schedule of Ctrl (*Kdm2a*^2lox/2lox^) and *Kdm2a* inducible knockout (iKO: *Kdm2a*^2lox/2lox^; *Actb*^CreERT2/+^) animals. **C** Testes of Ctrl and *Kdm2a* iKO males at 65 d post first tamoxifen treatment. Scale bar = 2 mm. **D** Combined testes weights of Ctrl and iKO mice at 65 d post first tamoxifen treatment. *n* = 6 males per group. Two-sample, two-sided Wilcoxon tests were performed between indicated groups ***P ≤ 0.001. Center lines in box plots represent median values, while lower and upper lines indicate the interquartile range (IQR; from the 25th to 75th percentile). **E** *Kdm2a* genotyping results of total testes genomic DNA showing progressive loss of the *Kdm2a*^2lox^ allele over time upon tamoxifen administration. (*n* = 1 animal per timepoint). However, at least three animals showed the maximal

biological effect (complete excision of exon 8 at day 15). **F** Immunofluorescence staining of testicular sections from Ctrl (*Kdm2a*^2lox/2lox^) (*n* = 1) and *Kdm2a* inducible knockout (iKO: *Kdm2a*^2lox/2lox^; *CAGG-Cre-ER^TM^*) (*n* = 1) at 9 d post first tamoxifen treatment with anti-c-KIT (green) and anti-KDM2A (red) antibodies. Acrosomes and nuclei were stained with Lectin PNA (blue) and DAPI (white), respectively. Seminiferous tubule borders are highlighted with yellow dashed lines. Yellow asterisk in interstitial space marks c-KIT-positive Leydig cells. The yellow arrowhead points to a KDM2A-positive SgU lacking c-KIT expression. Scale bar = 50 μm. **G** Scheme adapted from[2] and (74) summarizing expression and localization of KDM2A protein (red) in relation to developmental progression of male germ cells (related to Supplementary Fig. 2b). **H** PAS-stained testis section of Ctrl (stage VIII tubule) (*n* = 6) and *Kdm2a* iKO (*n* = 6) males, 65 d post first tamoxifen treatment. Scale bar = 50 μm.

chromosome synapsis, exchange of genetic information by recombination, and two meiotic divisions to form haploid spermatozoa.

Histone methylation serves key functions in gene regulation and development. Local levels of histone methylation are dynamically regulated by families of enzymes with opposing catalytic activities, including the relatively well-characterized histone methyltransferases and the less well-understood histone demethylases, that deposit and remove the modifications, respectively. The first histone lysine demethylase (KDM) identified belongs to the flavin-dependent monoamine oxidase family[5], while a second, larger family of demethylases, containing the Jumonji catalytic (JmjC) domain, was later identified[6]. KDM2A (also known as JHDM1A, FBXL11, or CXXC8) is the first characterized JmjC-domain containing KDM[7]. KDM2A has a paralog in mammals, called KDM2B (or JHDM1B, FBXL10, or CXXC2)[7]. Both enzymes predominantly demethylate the histone H3 lysine 36 dimethyl (H3K36me2) and, to a lesser extent, H3K36me1 modifications[7]. In addition to their highly similar catalytic domains (84% identical), KDM2A and KDM2B contain several conserved motifs, including a CXXC Zn-finger domain that specifically binds DNA containing non-methylated CpG dinucleotides[8,9]. Studies in mouse embryonic stem cells (ESCs) showed that most CpG islands (CGIs) are almost entirely devoid of DNA methylation and are bound by KDM2A and/or KDM2B. In contrast, CpGs outside of CGIs are highly methylated and depleted for KDM2A or KDM2B binding[8–10]. Recruitment of KDM2A/KDM2B to CGIs results in depletion of the highly abundant H3K36me2 mark, generating a chromatin architecture that is different from bulk non-CGI chromatin[8,9]. In mouse growing oocytes, KDM2A and KDM2B prevent DNMT3A-catalyzed de novo DNA methylation and recruit variant Polycomb Repressive Complex 1 (vPRC1) at CGI-gene promoters, thereby controlling H2A mono-ubiquitin deposition, vPRC1-dependent gene repression as well as downstream-acting PRC2-mediated H3K27me3 deposition[11]. Biochemically, H3K36 methylation inhibits the catalytic activity of the PRC2 complex[12]. Hence, by demethylating H3K36me2 on the H3 histone tail, KDM2A and/or KDM2B may promote PRC2 catalysis and gene repression.

KDM2A knockout mice exhibit reduced cellular proliferation, increased apoptosis, and severe developmental defects, resulting in mid-gestational embryonic lethality[13]. In order to bypass this lethality and to clarify the in vivo role of this gene in adult mammalian physiology and the regulation of transcription, we employed conditional mouse models in which *Kdm2a* was deleted in adult animals. We characterize KDM2A as a H3K36me2 demethylase active in spermatogonia, which is critical for mouse spermatogenesis. We identify previously unrecognized roles for the enzyme in the repression of genes controlled by Polycomb group proteins during spermatogenesis and in controlling X-linked gene expression during meiosis.

## Results

### Spermatogenesis is defective in *Kdm2a*-deficient mice

We sought to study the role of the H3K36me1/2 demethylase KDM2A[7] in adult mammal biology. To this end, we targeted the *Kdm2a* locus in C57BL/6 J ESCs with a vector containing an IRES-β-geo selection cassette and 3 loxP sites, two of which flank exon 8 (resulting in the *Kdm2a*^3lox^ allele; Fig. 1a). Deletion of this 94-bp exon removes a part of the catalytic center of the JmjC domain and results in a frameshift, leading to a functionally null allele. *Kdm2a*^3lox/+^ mice were mated with mice expressing the FLPe recombinase to create the conditional "floxed" *Kdm2a*^2lox^ allele. We then crossed *Kdm2a*^2lox/+^ mice with different transgenic Cre driver mice to investigate the consequences of temporally controlled or tissue-specific deletion of exon 8 of *Kdm2a* (the *Kdm2a*^1lox^ allele). Homozygous *Kdm2a*^1lox/1lox^ mutant mice died at mid-gestation as described previously in a comparable mouse model (deletion of Kdm2a exon 7–9[13]). To bypass the embryonic lethality, we generated *Kdm2a*^2lox/2lox^; *Actb*^CreERT2/+^ mice expressing a tamoxifen-inducible CreER^T2^ recombinase controlled by the endogenous β-actin promoter. Tamoxifen treatment induces widespread *Kdm2a* deletion in these mice (henceforth referred to as inducible knockout, or iKO, mice; Fig. 1a).

To determine the effect of *Kdm2a* loss in adult mice, we administered tamoxifen on five sequential days to twelve 10-week old iKO mice and twelve control (Ctrl) littermates (genotype *Kdm2a*^2lox/2lox^) equally distributed between genders, and carefully monitored them for 65 days (Fig. 1b). Adult iKO mice were viable and healthy, although males gained less body weight than Ctrl following tamoxifen administration (Supplementary Fig. 1a–c). At the end of the study, we performed a comprehensive post-mortem evaluation, including a histological characterization of over 40 tissues. The only apparent phenotype we observed in *Kdm2a* iKO mice involved the testes, which were markedly smaller and displayed an approximately 80% decrease in absolute weight (Fig. 1c, d). Within three days of the initiation of tamoxifen administration, partial recombination of the *Kdm2a* locus in the testes was observed by semi-quantitative PCR (Fig. 1e). This further increased at six days, and by nine days only residual levels of the floxed allele remained (Fig. 1e). RNA-sequencing (RNA-seq) analysis confirmed the almost complete removal of exon 8 in purified germ cells within 10 days post tamoxifen administration (see below).

To examine the expression pattern of *Kdm2a* and its paralogue *Kdm2b* during spermatogenesis, we reanalyzed previously published single-cell RNA sequencing data from adult mouse testes[14] (Supplementary Fig. 2a). *Kdm2a* and *Kdm2b* are highly expressed in a large fraction of premeiotic spermatogonia and at lower levels from pachytene spermatocytes onwards. In addition, *Kdm2a* expression was detected in somatic Sertoli cells. Immunofluorescence (IF) analysis of testicular sections showed that the KDM2A protein labels nuclei of undifferentiated spermatogonia strongly, and nuclei of differentiating spermatogonia, and of leptotene and zygotene spermatocytes more moderately (Fig. 1f, and Supplementary Fig. 2b). In nuclei of pachytene

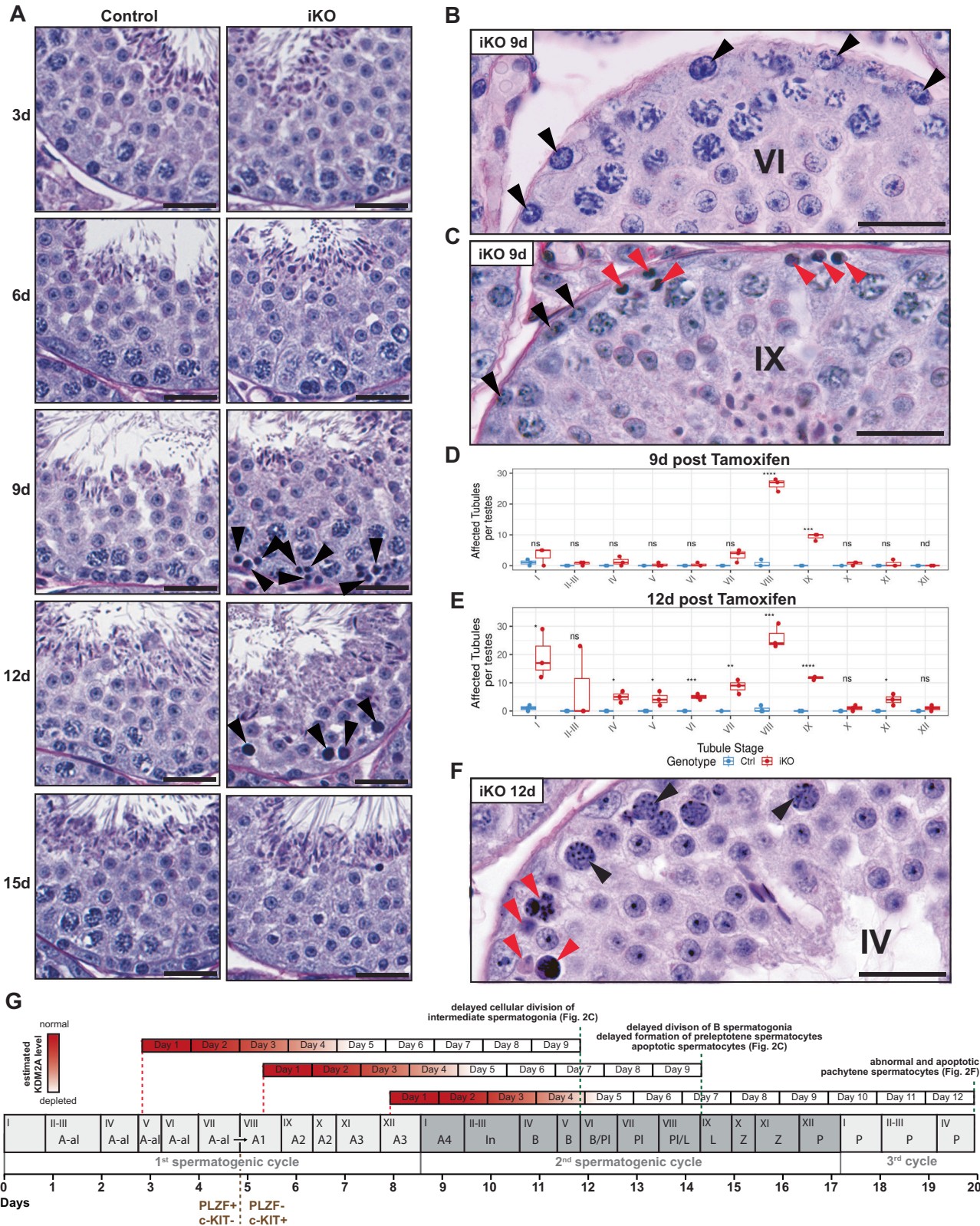

and diplotene spermatocytes and of early round spermatids, KDM2A becomes enriched at DAPI bright regions containing pericentromeric major satellite sequences packaged in a constitutive heterochromatin state (Fig. 1f, g and Supplementary Fig. 2b)[15]. After 9 days following tamoxifen treatment, all these germ cell populations exhibited greatly reduced levels of KDM2A in *Kdm2a* iKO testes (Fig. 1f and Supplementary Fig. 2b).

Histologically, 65 days after tamoxifen administration the *Kdm2a* iKO testes exhibited an almost complete loss of germ cells, with Sertoli cells and some spermatogonia remaining in the seminiferous tubules (Fig. 1h), as confirmed by immunohistochemistry for the TRA98 germ cell marker (Supplementary Fig. 1e). There was a corresponding absence of mature spermatozoa in the epididymis (Supplementary Fig. 1f). Tamoxifen administration to heterozygous floxed animals

**Fig. 2 | Defective spermatogenesis in *Kdm2a* iKO mice. A** PAS-stained section showing stage VIII seminiferous tubules of Ctrl and *Kdm2a* iKO males at 3, 6, 9, 12 and 15 d post first tamoxifen treatment. Arrowheads indicate apoptotic cells. Scale bar = 25 μm; *n* = 1 for 3 d and 6 d; *n* = 3 for subsequent time points. **B** Stage VI tubule of iKO male at 9 d (*n* = 3) containing Intermediate spermatogonia (arrowheads), indicative of delayed proliferation. Scale bar = 35 μm. **C** Stage IX tubule of iKO male at 9 d (*n* = 3) containing retarded preleptotenes (black arrowheads) and apoptotic spermatocytes (red arrowheads). Scale bar = 35 μm. **D**, **E** Stage-specific percentages of seminiferous tubules with apoptotic cells near the basal membrane at 9 d (*n* = 3) (**D**) and 12 d (*n* = 3) (**E**). We examined two testis cross sections per animal (373 to 573 tubules per testis). *n* = 3 per genotype, per time point. Two-sample, two-sided Wilcoxon tests were performed between the indicated groups. (nd) non determined, (ns) *P* > 0.05; (*) *P* < 0.05; (**) *P* < 0.01; (***) *P* < 0.001; (****) *P* < 0.0001. Center lines in box plots represent median values, while lower and upper lines indicate the interquartile range (IQR; from the 25th to 75th percentile). **F** Stage IV tubule of iKO male at 12 d (*n* = 3) showing morphologically abnormal (black arrowheads) and apoptotic (red arrowheads) pachytene spermatocytes. Scale bar = 35 μm. **G** Scheme summarizing aberrant phenotypes in *Kdm2a* iKO testes in relation to the timing of loss of KDM2A protein expression and of developmental progression of male germ cells (as determined by staging of seminiferous tubules).

(*Kdm2a*$^{2lox/+}$; *Actb*$^{CreERT2/+}$) resulted in no significant difference in the overall body weight (Supplementary Fig. 1d) nor testicular morphology (Supplementary Fig. 1g) compared to Ctrl males (*Kdm2a*$^{2lox/+}$). These data demonstrate that the loss of *Kdm2a* leads to a major impairment of male germ cell development.

### *Kdm2a*-deficient spermatogonia and spermatocytes undergo apoptosis

The presence of residual spermatogonia in the *Kdm2a* iKO testes at day 65 post-tamoxifen dosing suggested that the defect arose during the differentiation of spermatogonia. To study this process, we conducted a 15-day time course experiment, collecting testes every 3 days following the start of tamoxifen administration, which encompasses germ cell development over almost two cycles of the seminiferous epithelium[3]. Whereas testes from Ctrl and *Kdm2a* iKO males harvested at day 3 (3 d) and 6 d were histologically indistinguishable, iKO-derived testes obtained at 9 d exhibited marked degenerative changes, particularly at stage VIII, near the basal lamina of seminiferous tubules, including pyknosis and loss of germ cells. In contrast, spermatocytes and spermatids that are developmentally more advanced were unaffected, enabling accurate staging of germ cell progression (Fig. 2a, and Supplementary Fig. 3a). However, intermediate (In) spermatogonia, which normally divide and differentiate into B spermatogonia in stage IV, were instead dividing in stages V-VI (Fig. 2b). Moreover, B spermatogonia that normally divide in stage VI to form preleptotene spermatocytes exhibited delayed mitosis as well, dividing one or two stages later. In stage IX tubules, some remaining spermatogonia and many apoptotic spermatocytes were visible (Fig. 2c). Quantification indicated that degenerative defects manifested predominantly in stages VIII and IX (Fig. 2d). Given the approximate 4-5 days lag time in efficient *Kdm2a* deletion (Fig. 1e), these findings suggest that KDM2A depletion in differentiating spermatogonia delays cell cycle progression and reduces survival (Fig. 2g).

In 12 d samples, spermatogonial numbers were further reduced, and spermatocytes, up to the late-pachytene population in stage VIII, were morphologically abnormal or absent (Fig. 2a). Quantification revealed substantial degeneration of spermatocytes over multiple stages (Fig. 2e), including stages IV to VI (Fig. 2f), indicating that pachytene spermatocytes are also sensitive to *Kdm2a* deficiency, beyond the cell death observed for spermatogonia in 9 d samples. These stage IV-VI pachytene spermatocytes undergoing cell death likely had experienced KDM2A depletion from the preleptotene stage onwards, though *Kdm2a* deletion had been initiated before when they were approximately mid-differentiating type A3-A4 spermatogonia. Thus, these data argue that differentiating spermatogonia as well as spermatocytes are highly susceptible to KDM2A depletion, ultimately affecting their viability.

We next performed anti-cleaved caspase 3 (CC3) immuno-staining for all time points to identify cells undergoing apoptosis. Confirming the conclusions from the histological PAS-stainings, we observed CC3-signal near the basal membrane of the seminiferous tubules, predominantly in stage VII-IX tubules in 9 d iKO samples (Supplementary Fig. 3b). Quantifying the relative pixel area of CC3-staining in whole testis cross sections indicated a 4.5- and 4-fold increase in CC3-signal in the 9 d and 12 d iKO samples, respectively (Supplementary Fig. 3c).

### Depletion of c-KIT-positive differentiating spermatogonia in *Kdm2a* iKO mice

To investigate in more detail the fate of differentiating as well as undifferentiated spermatogonia upon *Kdm2a* gene disruption, we stained testis sections from 9 d and 12 d animals with antibodies marking spermatogonia at distinct developmental stages (indicated in Fig. 2g). Undifferentiated spermatogonia (As, Apr and Aal) are characterized by expression of the promyelocytic leukemia zinc finger protein PLZF/ZBTB16, whereas differentiating spermatogonia (A1-A4, In, and B) express the tyrosine-protein kinase c-KIT[1,16]. Immunohistochemistry using an anti-PLZF antibody showed comparable numbers of PLZF-positive undifferentiated spermatogonia in Ctrl and iKO testes at 9 d and 12 d (Fig. 3a, b). In contrast, the number of c-KIT-positive differentiating spermatogonia was reduced by 60% in *Kdm2a* iKO at 9 d, and the c-KIT-positive cells were nearly absent at 12 d (Fig. 3c, d). Importantly, at 65 d post-tamoxifen administration, the number of PLZF-positive cells (normalized to Sox9-positive Sertoli cells) remained identical in tubules of iKO mice and Ctrl animals (Fig. 3e, f). These results suggest that KDM2A function is dispensable for the maintenance of undifferentiated spermatogonia but is vital in c-KIT-positive differentiating spermatogonia.

### Spermatogonia-specific *Kdm2a* deletion causes testicular atrophy

To confirm that the impairment of male germ cell development was directly related to the loss of *Kdm2a* in spermatogonia, we introduced the *Ngn3-Cre* transgene, driving Cre recombinase expression in undifferentiated spermatogonia[17], into the *Kdm2a*$^{2lox/2lox}$ strain (henceforth referred to as spermatogonial knockout, sKO). Loss of *Kdm2a* in sKO resulted in severe testicular atrophy at 26 d of age (Supplementary Fig. 4a), with a 72% decrease in testes to body weight ratio compared to *Kdm2a*$^{2lox/+}$; *Ngn3-Cre* Ctrl (Supplementary Fig. 4b). Similar to 65 d iKO animals (Fig. 1f), seminiferous tubules of sKO males had a complete absence of spermatocytes and spermatids (Supplementary Fig. 4c) as well as a significant reduction of c-KIT-positive spermatogonia (Supplementary Fig. 4d, e). The number of PLZF-positive undifferentiated spermatogonia, calculated in relation to SOX9-positive Sertoli cells, was unaltered in the sKO males (Supplementary Fig. 4f–h), similar to 65 d iKO males (Fig. 3e, f). These results confirm that *Kdm2a* is not required for spermatogonial stem cell maintenance but rather is indispensable for the generation and maintenance of differentiating spermatogonia.

### Genome-wide transcriptional changes in c-KIT-positive differentiating spermatogonia in *Kdm2a* iKO

To understand the molecular role of KDM2A in driving spermatogonial differentiation and early meiosis, we analyzed genome-wide expression in purified cell populations from distinct developmental stages. To this end, we modified a FACS-based spermatocyte purification approach[18] by including an immuno-affinity step to isolate

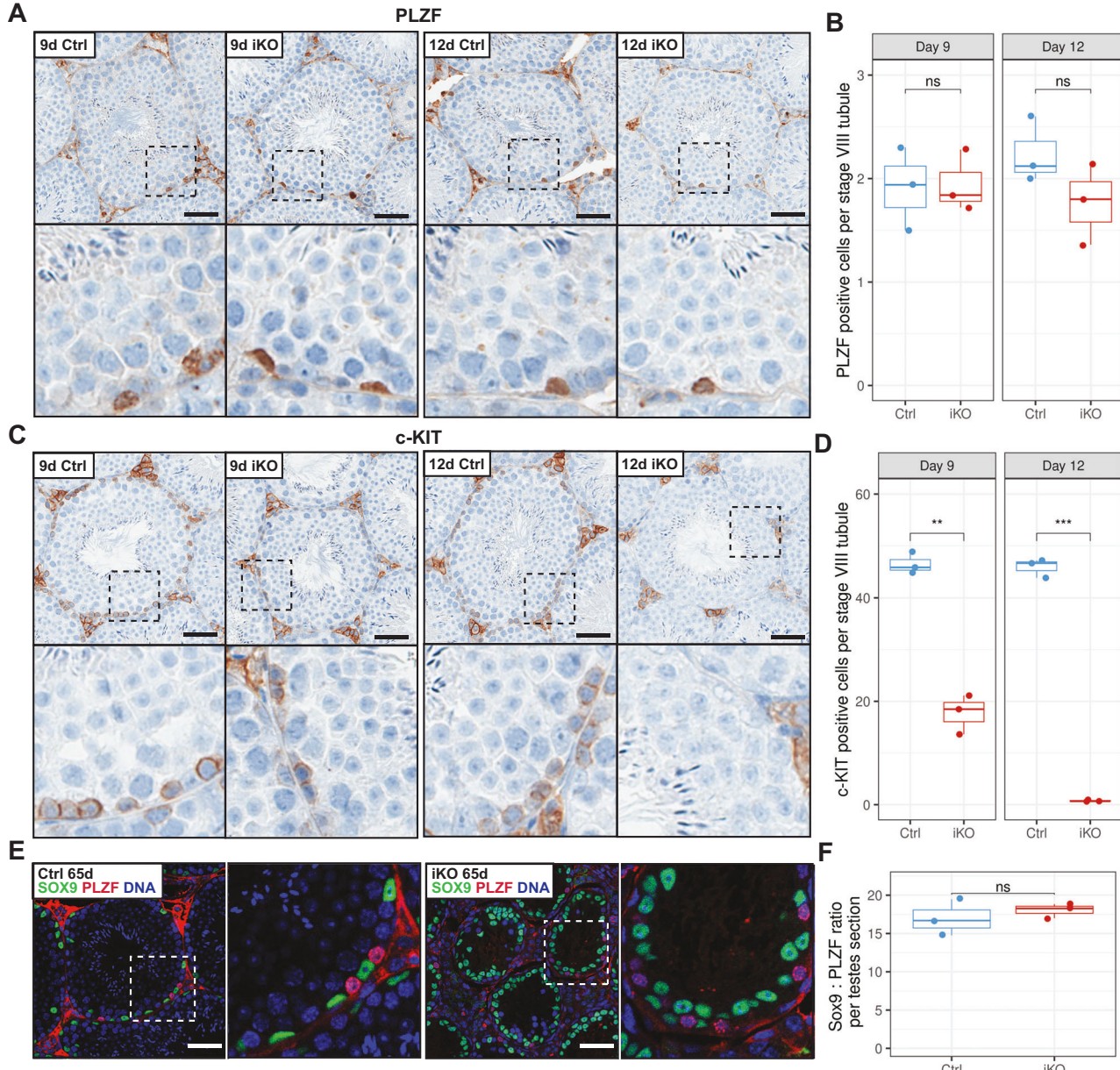

**Fig. 3 | Depletion of c-KIT + differentiating spermatogonia in *Kdm2a* iKO.**
**A**, **C** Immunohistochemical staining of 9 d and 12 d testes from Ctrl and *Kdm2a* iKO males with an anti-PLZF antibody (**A**), an anti-c-KIT antibody (**C**) and a hematoxylin counter stain. Boxed regions in the upper panels are magnified below. Note that Leydig cells outside the seminiferous tubules stain positive for c-KIT as well. Scale bar = 50 μm. **B**, **D** Quantification of PLZF + (B) or c-KIT + (**D**) cells in Ctrl and *Kdm2a* iKO seminiferous tubules at 9 d and 12 d. Fifty stage VIII seminiferous tubules were evaluated per mouse. $n$ = 3 per genotype and time point. Two-sample, two-sided Wilcoxon tests were performed between indicated groups. (ns) $P > 0.05$; (**) $P < 0.01$; (***) $P < 0.001$; (****) $P < 0.0001$. Center lines in box plots represent median

values, while lower and upper lines indicate the interquartile range (IQR; from the 25th to 75th percentile). **E** Immunofluorescence-staining of testes from Ctrl and *Kdm2a* iKO males at 65 d using antibodies against PLZF (red) and SOX9 (Sertoli cell marker, green), along with DAPI (Blue). Boxed regions in the left panels are magnified to the right. Scale bar = 50 μm. **F** Ratio of SOX9 + and PLZF + cells per testis cross-section. Over fifty seminiferous tubules per mouse were analyzed. $n$ = 3 males per genotype. Two-sample, two-sided Wilcoxon tests were performed between indicated groups. (ns) $P > 0.05$. Center lines in box plots represent median values, while lower and upper lines indicate the interquartile range (IQR; from the 25th to 75th percentile).

spermatogonia at successive stages of their development according to the expression of three cell surface markers: INTEGRIN ALPHA 6 (ITGA6)/CD49f, E-CADHERIN (CDH1/CD324) and the proto-oncogene protein c-KIT/CD117[19–21]. By sorting out cells high for both ITGA6 and CDH1 and low for c-KIT, we highly enriched for PLZF-positive spermatogonia (> 80%, $n$ = 46), which we hence refer to as undifferentiated spermatogonia (SgU). In contrast, 100% ($n$ = 75) of ITGA6$^{low}$/CDH1$^{low}$/c-KIT$^{high}$ cells were PLZF-negative, hereafter referred to as differentiated spermatogonia (SgD). In addition, we isolated early meiotic cells

enriched for leptotene and zygotene spermatocytes (ScLZ) by FACS (Fig. 4a; Supplementary Fig. 5a, b). The SgU, SgD, and ScLZ cell populations cover most of the spermatogenesis differentiation pathway, extending from the initial stages through to early meiosis.

For each genotype (Ctrl: *Kdm2a*$^{2lox/2lox}$ and *Kdm2a* inducible knockout iKO: *Kdm2a*$^{2lox/2lox}$; *CAGG-Cre-ER$^{TM}$*), we sorted out cells from 4 animals 10 d post-tamoxifen treatment. While the relative number of SgU and ScLZ cells were comparable, the SgD population was 2-fold reduced in *Kdm2a* iKO compared to Ctrl testis (Supplementary Fig. 5c).

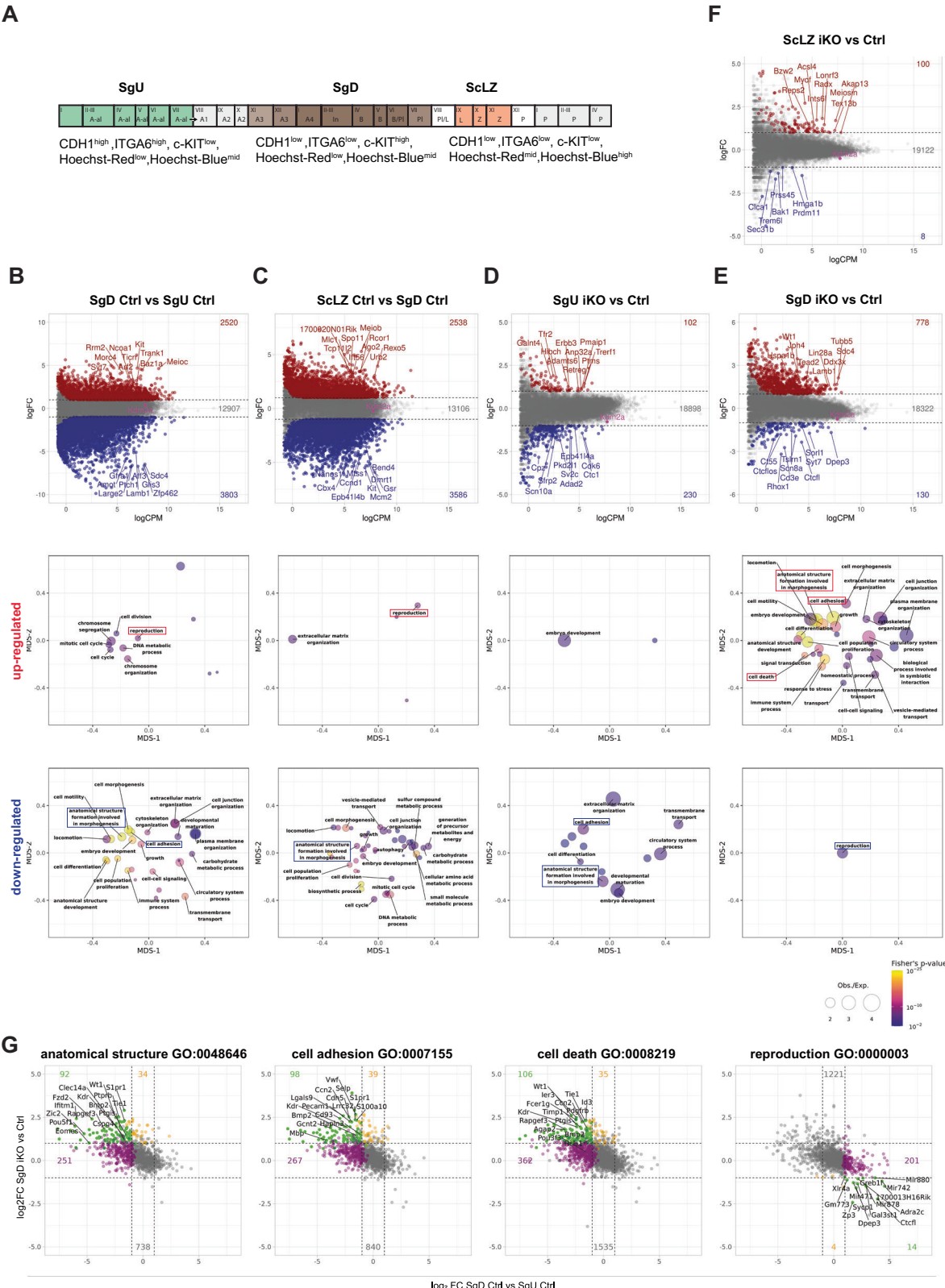

We purified mRNA from 10,000 cells per sample and performed bulk RNA-sequencing analyses. The total library size (22'530'688 – 42'003'250 reads) and the overall mapping of reads (61.5%–80.2%) were comparable between the various samples, and we observed high correlations between read distributions of replicates (Pearson correlation between 0.89 and 0.99). Principal component analysis showed that the primary variance in the dataset (48.02%) is captured by PC1,

which separates the different cell type groups and confirms that similar populations of germ cells existed in *Kdm2a* iKO mice as in Ctrl mice (Supplementary Fig. S6a). For the *Kdm2a* exons 1–7 and 9–21, we measured an approximately 35% reduction in the total read numbers in the iKO versus Ctrl samples (Supplementary Fig. 6b). For exon 8, floxed in the *iKO* model, we detected 5-fold reduction of sequencing reads in *Kdm2a* iKO germ cells, whereas we observed new splice junctions

**Fig. 4 | Kdm2a iKO mice fail to repress developmental genes during early spermatogenesis. A** Scheme showing FACS sorting antigens to isolate two populations of spermatogonia (Sg) (two differentiation stages: Undifferentiated (SgU) and Differentiated (SgD)) and early meiotic spermatocytes (Sc) (Leptotene/Zygotene (ScLZ)). **B–F** Scatter plots showing the gene expression fold-changes among **(B)** Ctrl SgD versus Ctrl SgU, **(C)** Ctrl ScLZ versus Ctrl SgD, **(D)** iKO SgU versus Ctrl SgU, **(E)** iKO SgD versus Ctrl SgD and **(F)** iKO ScLZ versus Ctrl ScLZ were plotted against average gene expression levels of the compared samples. We used 4 Ctrl ($Kdm2a^{2lox/2lox}$) and 4 $Kdm2a$ inducible knockout (iKO: $Kdm2a^{2lox/2lox}$; $CAGG$-$Cre$-$ER^{TM}$) at 10 d post first tamoxifen treatment. The number of genes significantly upregulated (red) or downregulated (blue) in the later developmental stage (**B** and **C**) or in the iKO (**D, E, F**) are displayed for each comparison (FDR < 0.05, $log_2FC > |1|$, likelihood ratio test within a negative binomial distribution). Names of the most significantly misregulated genes are shown for each comparison. Bubble plots showing significantly enriched GOslim terms associated with upregulated (upper) and downregulated (lower) genes in **(B)** Ctrl SgD versus Ctrl SgU, **(C)** Ctrl ScLZ versus Ctrl SgD, **(D)** iKO SgU versus Ctrl SgU, **(E)** iKO SgD versus Ctrl SgD and **(F)** iKO SgU versus Ctrl SgU. The size of bubbles denotes enrichment, and color denotes significance (Fischer's exact test). The distance of the bubble centers represents the similarity between genes contained in each GOslim term. GOslim terms appearing in all contrasts are highlighted with boxes. **G** Scatter plots showing the gene expression fold-changes among Ctrl SgD versus Ctrl SgU (x-axis) were plotted against the gene expression fold-changes among iKO SgD versus Ctrl SgD (y-axis) for genes belonging to GO slim terms: anatomical structure GO:0048646, cell adhesion GO:0007155, cell death GO:0008219 and reproduction GO:0000003. Purple highlighted genes change only in the developmental contrast (Ctrl SgD versus Ctrl SgU), orange highlighted genes change only in genotype contrast (iKO SgD versus Ctrl SgD), and green highlighted genes change in both developmental and genotype contrasts.

between exons 7 and 9 (Supplementary Fig. 6c–f), confirming the efficient deletion of exon 8.

Furthermore, we examined the expression of established markers for the different cell types. Whereas general germ cell markers like *Ddx4*, *Mael* and *Dazl* were highly expressed in all samples, markers for somatic Sertoli (including *Sox9*, *Amh* and *Fshr*) and Leydig (e.g., *Cyp11a1* and *Hsd3b1*) cells were barely expressed, allowing us to exclude substantial contamination by somatic cells (Supplementary Fig. S6g). Importantly, markers for undifferentiated spermatogonia like *Plzf*/*Zbtb16*, *Nanos3*, *Gfra1*, *Neurog3* and *Pou5f1* were expressed at significantly higher levels in the ITGA6$^{high}$/CDH1$^{high}$/c-KIT$^{low}$ SgU cells. The *Stra8, Meiosin* and the *H3f4* marker genes were highly expressed in ITGA6$^{low}$/CDH1$^{low}$/c-KIT$^{high}$ SgD cells (Supplementary Fig. S6g). Meiosis-related genes such as *Sycp3*, *Hormad1* and *Spo11* were highly expressed in the ScLZ spermatocytes, demonstrating that the three sorted populations represent genuinely distinct types of germ cells following the spermatogenesis differentiation path (Supplementary Fig. S6g).

We next analyzed global transcription profiles of the three germ cell populations in Ctrl and *Kdm2a* iKO mice. During normal development, we observed a balanced upregulation (2518) and downregulation (3802) of genes as spermatogonia progressed from the SgU to SgD stage (Fig. 4b). Subsequently, as differentiated spermatogonia cells progressed to early meiosis, we observed 2536 and 3584 genes being upregulated and downregulated respectively, in ScLZ compared to SgD (Fig. 4c). Gene ontology (GO) analyses showed prominent overrepresentation of "reproduction" functions for upregulated genes and of functions in "morphogenesis", "cell differentiation", "cell proliferation" and "cell adhesion" for downregulated genes, in both developmental contrasts (Fig. 4b, c).

When comparing *Kdm2a* iKO versus Ctrl, limited gene expression differences were observed for SgU and ScLZ cells (Fig. 4d, f). In contrast, in differentiating spermatogonia, SgD the iKO samples predominantly showed increased gene expression compared to Ctrl (779 genes) (Fig. 4e). This finding is in line with the histological observations at 9 days post tamoxifen treatment, in which undifferentiated spermatogonia and spermatocytes populations appear normal while differentiated spermatogonia showed degenerative defects. GO analyses revealed that genes upregulated in *Kdm2a* iKO SgD exhibited functions similar to those of genes normally downregulated during spermatogenic development (e.g., "morphogenesis" and "cell adhesion") (Fig. 4e and Supplementary Fig. 7a, b). In line, most genes that were upregulated in *Kdm2a* iKO SgD normally undergo repression during spermatogonial differentiation towards the c-KIT-positive stage (Fig. 4g). For example, genes encoding key transcription factors (*Pou5f1, Wt1, Zic2, Eomes, Lef1* and *Twist1*) and cell signaling proteins (*Fzd2, Kdr, Tgfbr2, Tgfbr3* and *Ret*) important for maintenance and differentiation of adult stem cells are highly expressed in Ctrl and iKO SgU and become subsequently suppressed only in Ctrl SgD, but not iKO SgD.

This indicates that *Kdm2a* is predominantly required for efficient gene repression during spermatogonial differentiation (Fig. 4g and Supplementary Fig. 7a, b). In addition, GO analyses identified an overrepresentation of genes with functions in "cell death" among genes upregulated in *Kdm2a* iKO SgD (such as *Tnf, Egfr* and *Myc*). Such upregulation might underlie the reduction of differentiating spermatogonia in *Kdm2a* iKO testes (Fig. 4e, 4g and Supplementary Fig. 7c). Conversely, GO analyses identified an overrepresentation of genes with functions in "reproduction" among genes downregulated in *Kdm2a* iKO SgD (Fig. 4e, g and Supplementary Fig. 7d).

## KDM2A is required for silencing of PRC2 target genes in differentiating spermatogonia

To mechanistically understand the KDM2A-dependent gene regulation, we performed a comprehensive chromatin analysis at gene transcription start sites (TSS) of FACS-sorted SgD cells from Ctrl and *Kdm2a* iKO ($Kdm2a^{2lox/2lox}$; $CAGG$-$Cre$-$ER^{TM}$ 10 days post-tamoxifen treatment) animals. To this end, we employed ultra-low input native ChIP-seq (ULI-NChIP-seq)[22] and generated genome-wide profiles of (a) H3K36me2, being the preferred enzymatic substrate of KDM2A that is typically widely distributed throughout the genome except for CpG islands (CGIs) where is underrepresented[7,23]; (b) H3K4me3, a modification which is commonly found at CGIs, to which KDM2A is recruited via its CXXC domain binding to CpG dinucleotides[8]; (c) H3K27me3, a modification preferentially deposited at CGIs by the Polycomb repressive complex 2 (PRC2) who's activity is inhibited by H3K36 methylation[12,24,25]; and (d) DNA methylation, as a downstream DNA modification mediated by the de novo DNMT3 enzymes that are recruited to chromatin by binding to H3K36me2 deposited by NSD1 in mouse male germ cells[26] (Supplementary Fig. 8a–e).

Hierarchical clustering of genes integrating their RNA expression along with chromatin modification (H3K36me2, H3K4me3 and H3K7me3) enrichments and DNA methylation level at their TSS resulted in nine gene clusters (Fig. 5a, first column). This clustering directly separated most genes located on autosomes in clusters (cl.) 1 to 8. Cluster 9 contains many X chromosome-linked genes, possibly due to the hemizygosity of X in males (Fig. 5a, second column). Genes in cl. 1 to 5 contained mostly genes with a CGI near their TSS, while genes in cl. 6 to 8 harbored CpG-poor TSS regions (Fig. 5a, third and fourth columns). Integration of KDM2A Cut&Run data from Feng and colleagues[27] revealed KDM2A enrichment in c-KIT-positive spermatogonia at CGI-positive promoters belonging to clusters 1–5 (Fig. 5a, c). This result is in agreement with its CXXC domain specificity and previous findings in ESCs[8].

Highly expressed genes throughout spermatogenesis were found in cl. 1 and cl. 8, associated with housekeeping and reproduction functions, respectively (Supplementary Fig. 9a) and showed minimal transcriptional changes between Ctrl and *Kdm2a* iKO cells (Fig. 5b). CGI-positive gene promoters in cl. 1 showed

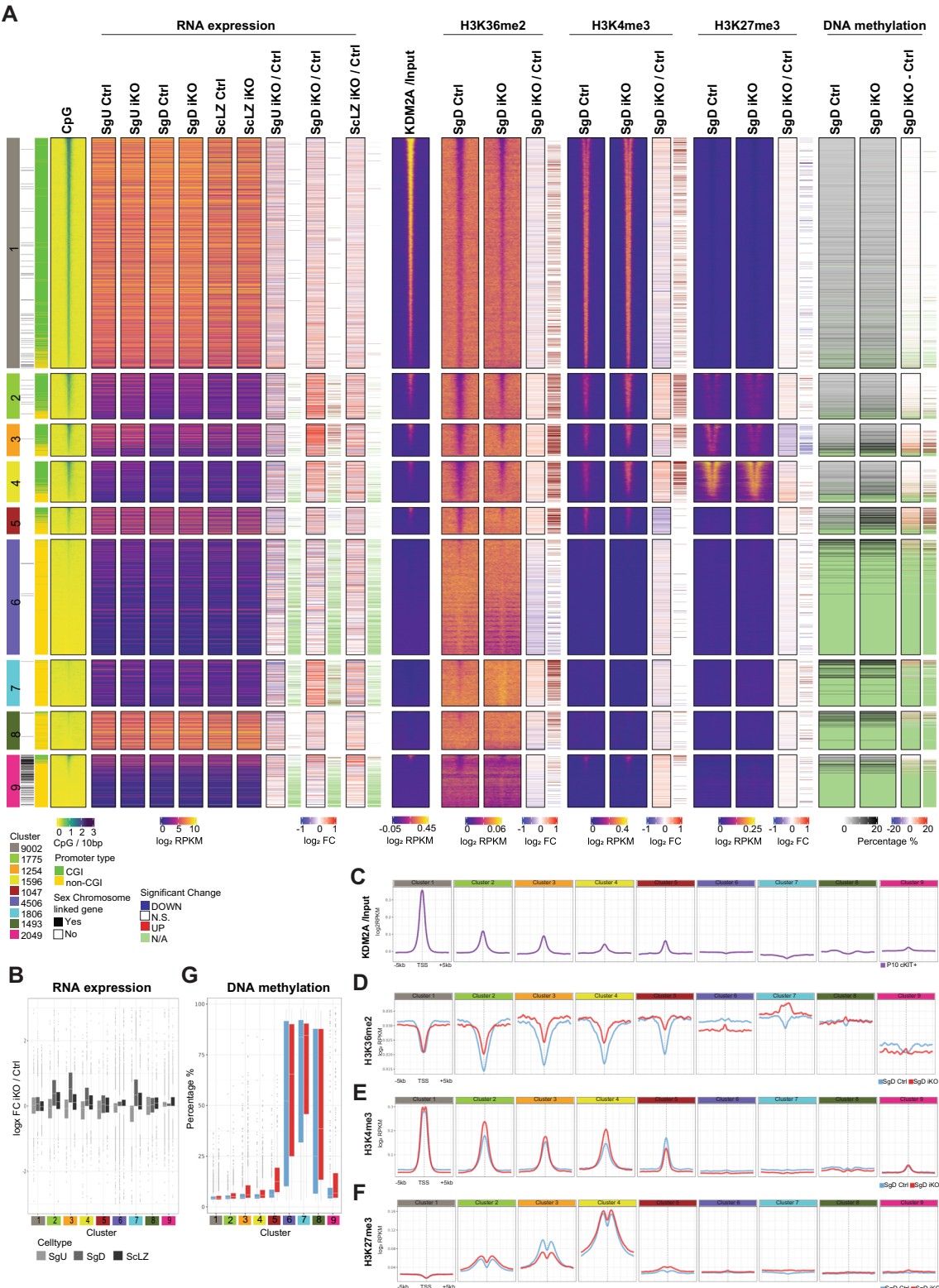

enrichment for KDM2A, depletion of H3K36me2 and high enrichment for H3K4me3 in both Ctrl and iKO SgD (Fig. 5a, 5c–e). Conversely, CGI-negative gene promoters in cl. 8 were devoid of KDM2A and H3K4me3 and exhibited higher levels of H3K36me2 compared to cl. 1 gene promoters in both Ctrl and *Kdm2a* iKO SgD cells. These data are in line with that H3K36me2 occupancy at promoters inversely correlates to local CpG density as reported previously in ESCs[23]. In addition, the anti-correlation between the KDM2A and H3K36me2 signal at CpG dense promoters (Supplementary Fig. 9c)

supports the proposed demethylase function of KDM2A's Jumonji domain[7]. Our data further argues that the gene and chromatin regulation of highly expressed genes is largely *Kdm2a* insensitive despite KDM2A binding to their promoters (Fig. 5a–c).

Genes in the CGI-rich promoter cl. 2 to 4 and in the CGI-poor promoter cl. 7 encompassed most genes upregulated in *Kdm2a* iKO SgD (Fig. 5a, b). They serve functions in "morphogenesis", "cell adhesion", "cell proliferation" and "cell differentiation" (Supplementary Fig. 9a, b).

**Fig. 5 | KDM2A target genes are associated with Polycomb repression.**
**A** Heatmap showing gene expression and chromatin signatures of 5 kb regions upstream/downstream of their corresponding TSS. Genes were classified into 9 clusters by hierarchical clustering using "Ward.D" agglomeration method on a dissimilarity matrix of "Euclidean" distance based on RNA expression values and chromatin modifications. From left to right: Cluster identity, Chromosomal location of the gene, Presence or absence of CpG island from the gene promoter, CpG density at TSS +/− 5 kb, $\log_2$ RPKM RNAseq values for SgU, SgD and ScZ samples from Ctrl ($Kdm2a^{2lox/2lox}$) and $Kdm2a$ iKO ($Kdm2a^{2lox/2lox}$; $CAGG$-$Cre$-$ER^{TM}$) at 10 days post initial tamoxifen treatment, $\log_2$ fold change (lFC) and statistical significance (FDR < 0.05 and lFC >|1|, likelihood ratio test within a negative binomial distribution) of transcriptional changes of genes between iKO versus Ctrl samples, $\log_2$ RPKM values of KDM2A minus input C&R (from[27]), and of H3K36me2, H3K4me3 and H3K27me3 ULI-NChIP-seq for non-overlapping 50 bp genomic windows located +/− 5 kb from the TSS in Ctrl and iKO SgD, $\log_2$ fold change and statistical significance (FDR < 0.05 and lFC >|1|) of chromatin modification enrichment changes of TSS +/− 1 kb between iKO versus Ctrl SgD samples, DNA methylation percentage assessed by RRBS at TSS +/− 1 kb in Ctrl and iKO SgD, DNA methylation

percentage difference and statistical significance (FDR < 0.05 and differential percentage >|20|) of TSS +/− 1 kb between iKO versus Ctrl SgD samples. **B** Boxplot showing RNA expression logFC for genes belonging to each cluster of $Kdm2a$ iKO over Ctrl SgU, SgD and ScLZ ($n = 4$ for from each genotype). Center lines in box plots represent median values, while lower and upper lines indicate interquartile range (IQR; from the 25th to 75th percentile). **C** Line plot showing average KDM2A $\log_2$RPKM values at TSS +/− 5 kb belonging to the gene clusters 1–9 as defined in Fig. 5, and assessed by C&R in wildtype cKIT-positive spermatogonia (from ref. 27). **D**–**F** Line plots showing average H3K36me2 (**D**), H3K4me3 (**E**) and H3K27me3 (**F**) $\log_2$RPKM values at TSS +/− 5 kb belonging to clusters 1–9 as defined in Fig. 5, and assessed by ULI-NChIP-seq from Ctrl ($Kdm2a^{2lox/2lox}$) and $Kdm2a$ iKO ($Kdm2a^{2lox/2lox}$; $CAGG$-$Cre$-$ER^{TM}$) animals 10 d post first tamoxifen treatment ($n = 2$ for H3K36me2 and H3K4me3 and $n = 4$ for H3K27me3 for each genotype). **G** Boxplot showing DNA methylation percentage of TSS +/− 1 kb belonging to each cluster of Ctrl ($n = 4$) and $Kdm2a$ iKO ($n = 4$) SgD samples. Center lines in box plots represent median values, while lower and upper lines indicate the interquartile range (IQR; from the 25th to 75th percentile).

CGI-poor promoter genes in cl. 6 and 7 were lowly expressed during spermatogenesis. They lack the gene activity-related H3K4me3 mark at their promoters. Some promoters are marked by DNA methylation (Fig. 5a, c). Particularly cl. 7 genes gained H3K36me2 at their promoters in $Kdm2a$ iKO compared to Ctrl SgD and showed transcriptional upregulation (Fig. 5a, b, d). Given the lack of KDM2A enrichment at promoters of cl. 7 genes, as measured by Cut&Run, the chromatin and transcriptional alterations in iKO SgD cells may either point to transient KDM2A binding and function at such non-CGI promoters, as seen in oocytes double deficient for $Kdm2a$ and $Kdm2b$[11] or represent secondary effects.

CGI-rich promoters in cl. 2 to 4 showed local H3K36me2 depletion (Fig. 5d), moderate enrichment for H3K4me3 and KDM2A (Fig. 5c, e) and low (cl. 2), moderate (cl. 3) and high (cl. 4) enrichment for H3K27me3 in Ctrl SgD (Fig. 5f). Importantly, genes in cl. 2 to 4 were lowly to moderately expressed in Ctrl SgU and particularly genes in cl. 2 and cl.3 became repressed in Ctrl SgD (Fig. 5a and Supplementary Fig. 9d). In $Kdm2a$ iKO SgD, however, gene promoters in cl. 2, 3 and 4 gained H3K36me2 (Fig. 5a, d), arguing that most of these CGI-containing promoters are subject to KDM2A-mediated demethylation (Fig. 5c). In addition, while gene promoters in cl. 2 to 4 showed increased H3K4me3 in $Kdm2a$ iKO compared to Ctrl SgD (Fig. 5a, e), only the gene promoters in cl. 3 showed reduced enrichment of H3K27me3 in $Kdm2a$ iKO versus Ctrl SgD. The latter change correlated with a failure to downregulate these genes during differentiation of iKO SgD compared to Ctrl SgD, (Fig. 5a, f, and Supplementary Fig. 9e, f).

Previously, it was shown that H3K27me3 and H2AK119ub1 occupancy showed dynamic changes across spermatogenesis[28,29]. To investigate the temporal dynamics of H3K27me3 occupancy during adult spermatogenesis, we generated genome-wide profiles in wildtype SgU, SgD, ScLZ and ScPD (Pachytene/Diplotene Spermatocytes) by ULI-NChIP-seq (Supplementary Figs. 8e, 10a, c, d). We further integrated the dynamics of PRC1-mediated H2AK119ub1 occupancy at gene promoters in wild-type type A spermatogonia during their early differentiation[28] (Supplementary Fig. 10b, c, e). Gene promoters in cl. 4 exhibited persistently high levels of H2AK119ub1 and H3K27me3 in all spermatogonial and spermatocyte cell types analyzed (Supplementary Fig. 10a, b, d, e). We observed an increase of H2AK119ub1 from undifferentiated to differentiated type A spermatogonia at both cl. 2 and cl. 3 gene promoters (Supplementary Fig. 10b, e). Notably, while H3K27me3 remained low at cl. 2 gene promoters in SgU, SgD, and ScLZ, and only increased during pachytene/diplotene of meiosis, H3K27me3 levels at cl. 3 gene promoters increased during the transition from SgU to SgD and continued to rise to pachytene spermatocytes (Supplementary Fig. 10a, d). The timing of such PRC1-catalyzed

H2AK119u1 and PRC2-catalyzed H3K27me3 deposition coincides with the marked repression of cl. 2 and cl. 3 genes in SgD cells (Fig. 5a and Supplementary Fig. 9d). Hence, these data indicate that $Kdm2a$ is required for the repression of a specific set of PRC1 and PRC2 target genes that undergo progressive gene silencing during spermatogonial expansion and differentiation.

## Impairment of meiotic progression to the pachytene stages in $Kdm2a$ iKO

In 9 d and 12 d histological preparations, we observed the presence of morphologically abnormal and apoptotic spermatocytes (Fig. 2). To understand this phenotype in greater detail, we investigated the entry of preleptotene spermatocytes into meiosis, which is normally characterized by major changes in gene expression, chromosome structure and dynamics. For example, upon completion of replication during pre-meiotic S-phase in preleptotene spermatocytes, programmed DNA double-strand breaks (DSB) form throughout the genome, facilitating homology-based invasion of single-stranded DNA sequences into homologous chromosomes and pairing of these chromosomes. This ultimately enables the exchange of genetic material between, and disjunction of, parental chromosomes in the two meiotic cell divisions. The formation and processing of DSBs is accompanied by extensive phosphorylation of S139 of H2A.X (γH2A.X) in chromatin around the DSB by the ATM kinase[30]. In parallel, proteinaceous fibers called axial elements form along chromosome axes in leptotene cells. In zygotene cells, progressive chromosome pairing leads to the formation of bipartite zipper-like structures between homologs, so-called "synaptonemal complexes" (SC)[31]. Spermatocytes enter the pachytene stage once all autosomal homologs complete synapsis. At the same time, γH2A.X levels at autosomes have been greatly reduced while the ATR kinase maintains H2A.X phosphorylation along the unsynapsed, non-homologous chromosome arms of the X and Y chromosomes, which are also marked by the synapsis surveillance protein HORMAD1[32,33].

In cryosections of 9 d testes and in drying-down preparations of individual spermatocytes[34], we observed comparable γH2A.X signals in leptotene and zygotene cells of iKO and Ctrl animals (Fig. 6a and Supplementary Fig. 11). Axial element and SC formation was invariant between Ctrl and iKO leptotene and zygotene spermatocytes (Fig. 6a and Supplementary Fig. 11). However, upon reaching the early pachytene stage, progression of meiotic prophase was impaired in iKO spermatocytes. Whereas autosomes in Ctrl spermatocytes had completed synapsis, and downregulated γH2A.X levels except on the sex chromosomes, iKO cells displayed prominent γH2A.X signals throughout their nuclei (Fig. 6a, b and Supplementary Fig. 11). The persistent γH2A.X signals in such pachytene-like cells correlated with a failure of autosomes to complete synapsis, as demonstrated by

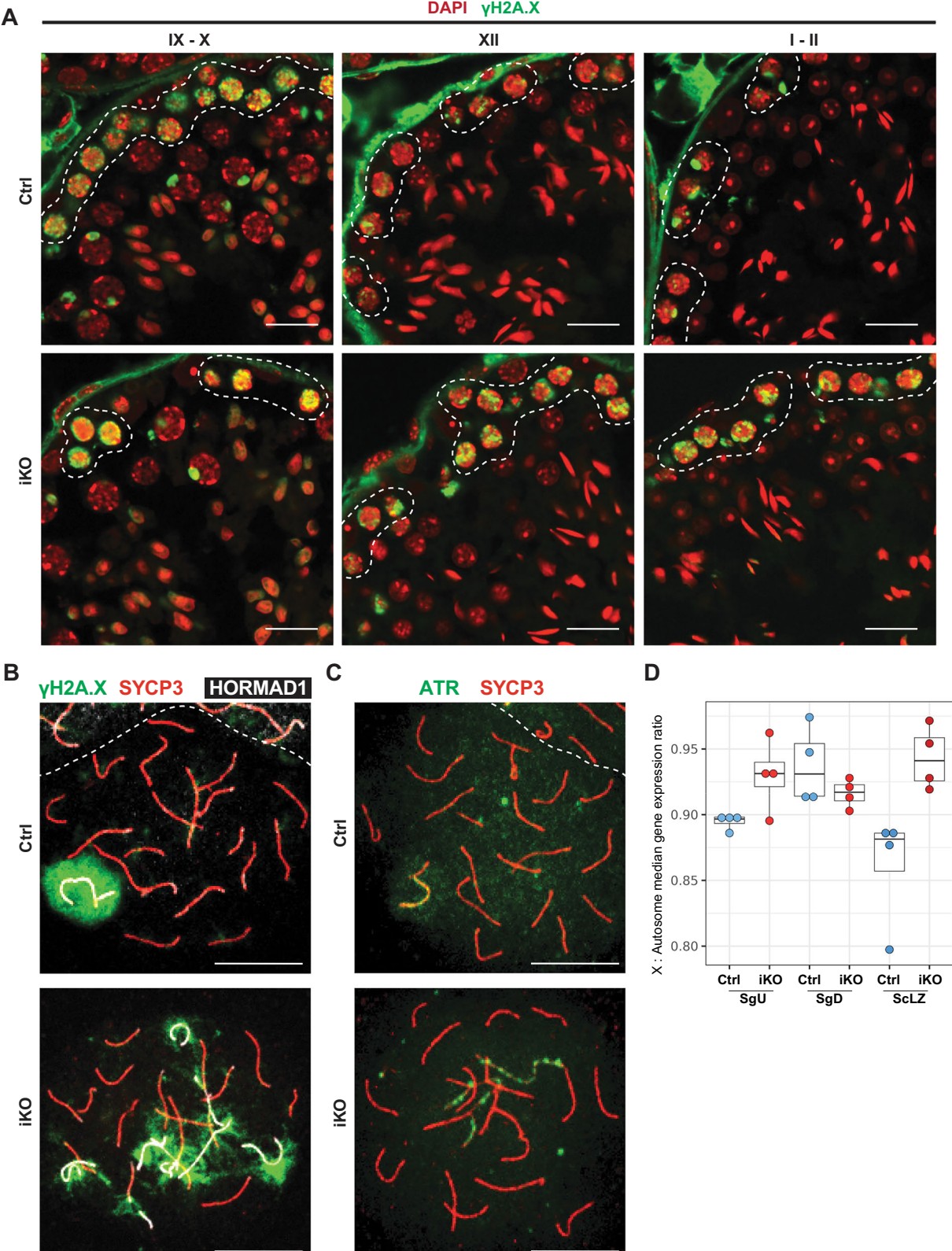

labeling of aberrant asynaptic chromosomes by the synapsis surveillance HORMAD1 protein and the ATR kinase (Fig. 6c and Supplementary Fig. 11). Thus, while *Kdm2a* deficient spermatocytes can initiate DSB formation and chromosome synapsis, they are defective in the processing of DSBs and/or completing synapsis.

We then compared the median expression ratio of X-linked versus autosomal genes, which allows us to track sex chromosome activity (Fig. 6d). We noticed that the X:A ratio in Ctrl ScLZ was decreased compared to the Ctrl spermatogonia (Fig. 6d). The downregulation of X-linked expression in Ctrl ScLZ cell population relative to SgD is notable since meiotic sex chromosome inactivation (MSCI) is thought to initiate silencing of X-linked gene expression in pachytene stage spermatocytes[35]. We further observed that the X:A ratio in *Kdm2a* iKO ScLZ samples remained similarly high as observed in the preceding

**Fig. 6 | Impairment of meiotic progression to the pachytene stages in *Kdm2a* iKO. A** Immunofluorescence staining of testicular sections from Ctrl (*Kdm2a*$^{2lox/2lox}$) and *Kdm2a* inducible knockout (iKO: *Kdm2a*$^{2lox/2lox}$; *Actb*$^{CreERT2/+}$) at 9 d (*n* = 3) post first tamoxifen treatment with an anti-γH2A.X antibody. Representative tubules in stage IX-X, stage XII, and stage I-II are shown, with the dashed lines highlighting early meiotic cells (leptotene (IX-X), zygotene (XI-XII), and early pachytene (I-III) stage, respectively). Scale bars = 10 μm. **B** Representative images of drying-down spermatocytes from Ctrl and iKO animals at 9 d (*n* = 3) post first tamoxifen treatment co-stained with anti-γH2A.X, anti-SYCP3, and anti-HORMAD1 antibodies. Scale bars = 10 μm. **C** Representative images of dried-down spermatocytes from Ctrl and iKO animals at 9 d post first tamoxifen treatment co-stained with anti-ATR and anti-SYCP3 antibodies. **D** Boxplots showing the ratio of the median of expression of X-linked genes divided by the median of expression of autosomal genes for 4 Ctrl (*Kdm2a*$^{2lox/2lox}$) and 4 *Kdm2a* inducible knockout (iKO: *Kdm2a*$^{2lox/2lox}$; *CAGG-Cre-ER*$^{TM}$) at 10 d post first tamoxifen treatment SgU, SgD and ScLZ cell types. Center lines in box plots represent median values, while lower and upper lines indicate the interquartile range (IQR; from the 25th to 75th percentile).

*Kdm2a* iKO spermatogonia (Fig. 6d). Accordingly, we detected higher expression of X-linked genes in *Kdm2a* iKO versus Ctrl ScLZ (Supplementary Fig. 12a, b). The failure to induce down-regulation of X-linked genes in *Kdm2a* iKO ScLZ population may point to a role of KDM2A in DNA damage response-induced gene silencing. This role is distinct from the *Kdm2a*-dependent autosomal PRC2-related gene repression that we had identified at the preceding stage of differentiated spermatogonia. Consistently, the H3K36me2, H3K4me3 and H3K27me3 histone marks were not or only marginally changed at promoters of X-linked genes in SgD samples (Supplementary Fig. 12a, c). Taken together, the inefficiency in the developmentally programmed down-regulation of X-linked gene expression in *Kdm2a* iKO ScLZ, along with the widespread retention of ATR and γH2A.X on chromatin in *Kdm2a* iKO pachytene-like spermatocytes, identifies KDM2A as a potentially direct or indirect regulator of DNA damage response-induced gene silencing[36].

Beyond defects directly related to meiosis, we also observed changes in the dynamics of histone modifications. Levels of H3K4me3 and H3K9me2 have been reported to be downregulated upon entry into the pachytene stage[37,38]. In Ctrl animals, we confirmed the down-regulation of these marks, whereas in iKO testes, both modifications remained elevated in stage I-II spermatocytes, at levels observed in leptotene and zygotene spermatocytes (Supplementary Fig. 13a, b). Conversely, global levels of H3K36me2, H3K36me3 and H3K27me3 appear comparable between Ctrl and iKO spermatocytes (Supplementary Fig. 14a–c). These data support the notion that *Kdm2a* deficiency during spermatogonial differentiation and early meiosis may impair multiple processes normally occurring during the progression of spermatocytes to the pachytene stage.

## Discussion

In this study, we identified an essential role for the histone demethylase KDM2A in spermatogenesis. Induction of *Kdm2a* deficiency in adult males impaired germ cells at multiple stages of their development, resulting in progressive loss of mature spermatozoa over time. Other tissues in adult mice remained, however, largely unaffected. Both in the tamoxifen inducible (iKO) and in the constitutive *Ngn3-Cre* driven (sKO) deletion models, we observed that differentiating spermatogonia were highly susceptible to *Kdm2a* deficiency, as reflected by reduced numbers of c-KIT-positive spermatogonia, delayed progression through the cell cycle of intermediate and B spermatogonia, and increased apoptosis resulting in severe spermatogenic defects and infertility. In contrast, we measured similar numbers of PLZF-positive spermatogonia between knock-out and Ctrl animals, indicating that *Kdm2a* is dispensable in self-renewing spermatogonia. The observed cell cycle defects in differentiating spermatogonia are reminiscent of defects observed in *Kdm2a*-deficient embryos, such as growth retardation, decreased proliferation and increased levels of apoptosis, culminating in embryonic lethality at E10.5–12.5[13]. These data demonstrate an important role of KDM2A in the control of cell cycle progression during embryonic and male germ cell development. In contrast, the development of *Kdm2a*-deficient oocytes is not grossly affected[11,39], illustrating a major sexual dimorphism in the function of KDM2A in both germ lines.

In this study, we identified over 3500 genes that become progressively repressed during spermatogonial differentiation. According to gene ontology analysis, these genes serve functions in multicellular organism development and morphogenesis. We identified *Kdm2a* as a key repressor of over 750 genes during this process. Intriguingly, despite a global gain of H3K36me2 at CGI-promoters in *Kdm2a*-mutant spermatogonia, when relating the transcriptional changes to other chromatin modification states, we found that transcriptionally, *Kdm2a* susceptible genes were generally marked by Polycomb-associated chromatin, which exhibited dynamic changes throughout spermatogonial transitions. In ESCs, KDM2A and its paralog KDM2B localize at CGIs, including promoters of CGI genes, due to the presence of the CGI-interacting CXXC motif in both proteins[8,9,40,41]. In these cells, KDM2B preferentially localizes to CGI-promoters marked by components and chromatin modifications of PRC1 and PRC2, while KDM2A is underrepresented at such genes[9,42]. *Kdm2b* is also expressed during spermatogenesis. In contrast to *Kdm2a* deficiency, however, loss of function of the *Kdm2b* long isoform, encoding for the JmjC-domain-containing enzyme, did not affect spermatogonial differentiation. Instead, spermatogonial stem cell maintenance appeared to be partially impaired upon increasing age[43]. Thus, the demethylase activities of the two enzymes are developmentally non-redundant and control different cellular processes during spermatogenesis. Given the impaired repression of canonical PcG target genes in iKO SgDs, we speculate that KDM2A contributes to developmentally related transcriptional repression by driving PRC1/PRC2-mediated gene repression.

For PRC2, conditional deficiency of *Eed* and *Suz12* affected synapsis of homologous chromosomes and telomere clustering during meiosis[44]. Contrasting with *Kdm2a* deficiency, however, loss of PRC2 proteins resulted in progressive reduction of PLZF⁺ spermatogonia during adulthood, suggesting defective spermatogonial stem cell maintenance and/ or differentiation[44]. Consistently, while global levels of H3K27me3 display a reduction during transition of undifferentiated to differentiated spermatogonia[28], multiple gene promoters which are moderately marked by H3K27me3 and lowly expressed in undifferentiated spermatogonia, gain H3K27me3 during differentiation of spermatogonia and during meiosis, suggesting dynamic control of PRC2 function during spermatogenesis (Supplementary Fig. 10a and ref. 29). Given the allosteric inhibition of PRC2 activity by H3K36me2[12,24,25], KDM2A may function upstream of PRC2 by reducing H3K36me2 levels, thereby facilitating PRC2 catalysis of H3K27me3.

Beyond controlling the catalysis of PRC2, our reanalysis of PRC1-catalyzed H2AK119u1[28] also showed a dynamic gain of this mark at cl.2 and cl.3 gene promoters upon differentiation of type A spermatogonia (Supplementary Fig. 10b, c). Subsequently, PRC2 function becomes enhanced at cl.2 genes in pachytene spermatocytes, while at cl.3 genes, it is already enhanced in SgD cells (Supplementary Fig. 10a, c). These findings are in line with the notion that PRC1 acts upstream of PRC2 in ESCs, establishing a H2AK119u1 chromatin state that subsequently drives PRC2 recruitment[42].

Given the transcriptional upregulation of cl.2 genes in SgD iKO cells, this raises the intriguing possibility that H2AK119ub1 deposition by PRC1 is also compromised in *Kdm2a*-mutant cells, akin to

H3K27me3 reduction observed at cl.3 genes. Hence, analogous to the recruitment of the variant PRC1.1 complex by KDM2B in ESCs[42,45], KDM2A may be also directly involved in the recruitment of possibly germ-line specific PRC1-like complexes to CGIs via its CXXC domain, thereby contributing to repression of PcG target genes in spermatogonia. A similar recruitment mechanism of PRC1 via KDM2A likely functions in growing oocytes as well[11].

Comparable to *Kdm2a*, the PRC1 subunit *Scml2* is required for spermatogonial differentiation and meiosis, as demonstrated by reduced numbers of STRA8[+] and SYCP3[+] cells, while spermatogonial stem cell maintenance was unaffected[46,47]. Based on transcriptional and chromatin analyses in control GS cells, as well as in purified THY1[+] spermatogonia, pachytene spermatocytes and round spermatids of mutant and control animals, Hasegawa and colleagues suggested that SCML2 promotes repression of PRC1 target genes by facilitating H2AK119u1[46], presumably during spermatogonial proliferation. Of note, depletion of both catalytic subunits of PRC1, *Ring1* and *Rnf2*, from undifferentiated spermatogonia leads to accelerated cell cycle and promotion of precocious differentiation[28]. Together, our study and published work show that germ-line mutants for *Kdm2a* and various components of PRC2 and PRC1 complexes share multiple phenotypes, and that these chromatin modifiers likely co-regulate similar gene expression programs that are essential for successful spermatogenesis.

Loss of KDM2B expression in ESCs resulted in aberrant DNA methylation at some CGIs controlled by PRC1[48]. In male and female germlines, de novo DNA methylation is mainly facilitated by DNMT3A, which potentially interacts with H3K36me2 through its PWWP domain. When NSD1 was depleted, H3K36me2 levels decreased, leading to a reduction in DNA methylation and an expansion of Polycomb-mediated gene silencing[26]. In *Kdm2a/Kdm2b* doubly deficient fully grown oocytes, CGI-promoters gained aberrant de novo DNAme, but only when H3K4me3 occupancy levels were low early during oocyte growth[11]. Contrary, in *Kdm2a*-mutant spermatogonia, CGI-rich promoters largely resisted abnormal gains of DNA methylation, possibly due to their generally high occupancy of H3K4me3, deposited by *Kmt2b/Mll2*[49], and which is known to counteract the binding of the de novo DNA methyltransferases[11,50]. Consistently, gene promoters in cl. 5 exhibiting the lowest H3K4me3 marking among CGI-rich gene promoters showed the highest gain of DNA methylation in iKO SgD cells. In addition, the presence of KDM2B in *Kdm2a*-deficient spermatogonia could sustain a basal level of H3K36 demethylation activity, which could prevent excessive gain of DNA methylation. Further investigation of the simultaneous depletion of *Kdm2a* and *Kdm2b* is required to shed light on the interplay of H3K36me2 and DNA methylation at gene promoters during spermatogenesis.

Finally, we measured notable changes in X-linked gene expression in Ctrl and *Kdm2a* iKO spermatocytes, as well as a developmental arrest of *Kdm2a* iKO spermatocytes occurring at the pachytene stage of meiotic prophase I. Previous cytological research on meiotic prophase substages identified significant lower levels of chromatin-associated RNA Polymerase II in leptotene and zygotene compared to pachytene and diplotene spermatocytes[51]. In line, an extensive RNA-seq study reported a transient global reduction in gene expression in preleptotene and leptotene spermatocytes compared to preceding spermatogonial and later stage spermatocyte populations[52]. We hypothesize that such transient genome-wide transcriptional down-regulation may result from BRCA1 and ATM signaling at DSBs and unsynapsed chromosome axes taking place in early meiotic cells[52,53]. Subsequently, as DSB repair and synapsis of autosomal chromosomes proceed from zygotene to pachytene stages, autosomal genes become reactivated, while X-linked genes become further repressed at pachynema and diplonema in a process known as meiotic sex chromosome inactivation (MSCI)[33,51,54]. In our study, FACS-sorted Ctrl ScLZ populations contain a mixture of early meiotic prophase I spermatocytes (10%

preleptotene, 40% leptotene, 40% zygotene and 10% early pachytene spermatocytes. In Ctrl ScLZ cells, we measured a reduction in the expression ratio of chromosome X-linked (X) genes versus autosomal (A) genes, as compared to spermatogonia (Fig. 6d). Following our hypothesis raised above, we propose that ensuing the transient global repression in leptotene spermatocytes, the autosomal genes in zygotene and early pachytene cells of the Ctrl ScLZ population became upregulated while X-linked expression did not, thus explaining the observed reduction in the X:A gene expression ratio.

Failure to complete autosomal chromosome synapsis by the end of the zygotene stage and retention of the DNA damage response factors at unsynapsed autosomes are thought to drive meiotic silencing of unsynapsed chromatin (MSUC), which, depending on its extent, may prevent subsequent MSCI[53–55]. While meiotic DSB formation and chromosome synapsis were initiated in *Kdm2a*-deficient spermatocytes, these processes failed to be completed at the developmental timing of the pachytene stage, resulting in persisting unsynapsed chromosome axes labeled by the synapsis surveillance protein HORMAD1, the ATR DNA damage kinase, and chromatin highly labeled with γH2A.X (Fig. 6a–c). Although MSCI is thought to begin in pachytene spermatocytes, we detected an upregulation of X-linked genes already in *Kdm2a* iKO ScLZ cells (Fig. 6d). Accordingly, the pachytene-like spermatocytes in *Kdm2a*-mutant mice failed to compartmentalize the X and Y chromosomes into a peripheral nuclear subdomain called the sex-body (Fig. 6b, c), further supporting that the X and Y chromosomes were not properly silenced. Mice bearing mutations in proteins associated with DNA damage response, such as BRCA1[56], MDC1[57] and H2A.X[57], also exhibit aberrant meiotic chromosome synapsis, MSCI failure and developmental arrest around the mid-pachytene stage, as we observed for *Kdm2a*-mutant spermatocytes. In human[58] and yeast[59] cells, di-methylation of H3K36 by SETMAR or MMSET at DSBs or telomeres, respectively, recruits Ku70 and NBS to promote non-homologous end joining (NHEJ). We hypothesize that during meiosis, KDM2A may be required to modulate H3K36me2 levels at DSBs to promote DNA damage repair via homologous recombination. Collectively, the persistence of meiotic pairing defects, the failure to complete meiotic DSB repair and to induce MSCI, possibly in conjunction with an incomplete maturation of meiotic chromatin, as exemplified by persistent elevated levels of H3K4me3 and H3K9me2, likely underlie the meiotic developmental arrest of *Kdm2a*-deficient pachytene-like spermatocytes.

In an initial study investigating KDM2A's role in spermatogenesis, Xiong et al.[60] generated germ cell-specific *Stra8*-Cre driven deletion of *Kdm2a* exon 6 and observed partial abnormalities in seminiferous tubules and oligospermia, concluding that KDM2A is nonessential for spermatogenesis. In contrast, a more recent study by Feng et al.[27], using *Stra8*-GFP-Cre and tamoxifen-induced *Ddx4*-CreERT2 deletion models also targeting exon 6 of *Kdm2a*, reported reduced testis weight, infertility, and defects in meiotic initiation and progression that was associated with reduced activation of meiotic gene expression in juvenile mice.

In agreement with Feng et al.[27], we independently established that KDM2A is required for male germ cell development. While Feng et al.[27] demonstrated that KDM2A interacts with HCFC1 and E2F1, to potentially regulate gene expression, meiotic entry and spermatocyte viability[61], our findings revealed defects in meiotic progression rather than initiation, which is likely associated with an impairment of MSCI. Contrasting Feng et al.[27], we further characterized KDM2A as a H3K36me2 demethylase controlling gene repression during spermatogonial differentiation in adult mice, showing an increase in H3K36me2 at H3K27me3-marked CGI-containing gene promoters. This was accompanied by defects in H3K27me3 deposition and gene repression, linking KDM2A's catalytic H3K36me2 demethylase activity to PRC2-mediated repression of target genes during spermatogonial differentiation. The differing conclusions regarding KDM2A's catalytic

activity and function across developmental stages, as seen in our work and Feng et al.[27], likely arise from variations in the timing of Cre recombinase activation, sample collection, differing chromatin profiling protocols and, critically, genomic data normalization procedures. Feng et al. primarily focused on *Stra8*-Cre-mediated *Kdm2a* excision during the first wave of spermatogenesis in juvenile mice, which highlighted defects in early meiotic progression. In contrast, our study employed a more extensive time-course approach during adult spermatogenesis, identifying differentiating spermatogonia as the earliest population of male germ cells that are sensitive to the loss of the H3K36me2 demethylase function of KDM2A. This distinction underscores the importance of KDM2A function across multiple developmental windows during spermatogenesis.

## Methods

### Generation of *Kdm2a* mutant mice

All animal procedures employed in this study were approved by the Novartis Institutes for BioMedical Research Institutional Animal Care and Use Committee (Protocol Numbers: 11 DP 037 and 14 DMP 049), conformed to the Swiss Animal Protection Ordinance (protocol number: 2612) and are compliant with FMI institutional guidelines. The *Kdm2a* gene targeting vector, in which a ~ 0.5-kb region containing exon 8 (94 bp) is flanked by *loxP* sites, was generated by sequentially cloning *Kdm2a* genomic fragments (5′ homology arm, 3′ homology arm, and floxed region), as well as an IRES-βgeo selection cassette flanked by FRT sites, into pBluscript II SK (Stratagene). The *Kdm2a* genomic fragments were generated by PCR using mouse genomic DNA as the template and the following pairs of primers: 5′-ATTGCGGCCGCTGTAGTATGTCACAGTCAC-3′ and 5′-AGAACTAGTGCTATGTAATTTCTCACATAC-3′ (5′ arm, ~ 4.4 kb, restriction sites used for cloning are underlined); 5′-GTCAGTCGACCAATATTTGGCCAGTTC-3′ and 5′-CTAGGCCGGCCAGGCTTCCCTTAAACTC-3′ (3′ arm, ~ 3.0 kb); 5′-ATTGGCGCGCCTGGAAATGTGACATAGAAC-3′ and 5′-TAACTCGAGTTGAATAGCTACAAAAGGT-3′ (floxed region, ~ 0.5 kb). All primers were purchased from Integrated DNA Technologies, Inc. (Coralville, Iowa, USA). The targeting vector was verified by DNA sequencing.

C57BL/6 J ESCs were electroporated with the *Kdm2a* targeting vector and then selected with G418. Antibiotic-resistant clones were initially screened for targeted integration by long-range PCR, and positive clones were confirmed by Southern blot. Correctly targeted (*Kdm2a*^3lox/+^) ESCs were injected into C57BL/6 J blastocysts to generate chimeric mice. *Kdm2a*^3lox/+^ animals were crossed with mice expressing FLPe in the germline to remove the IRES-βgeo cassette, thereby generating the conditional *Kdm2a*^2lox^ allele. *Kdm2a*^2lox/+^ mice were then crossed with mouse strains containing either CreER^T2^ expressed from the endogenous *Actb* locus[62] or CreER^TM^ expressed from the CMV-IE enhancer/chicken β-actin/rabbit β-globin hybrid promoter[63] (The Jackson Laboratory; JAX stock number: 004682), which give widespread expression of Cre that is active in the presence of tamoxifen, or with a constitutively active Cre driven by the *Ngn3* promoter (The Jackson Laboratory; JAX stock number: 006333), which expresses Cre in spermatogonia.

Activation of the *Actb*^CreERT2^ or *CAGG-Cre-ER*^TM^ was achieved by tamoxifen administration (Sigma-Aldrich; 1 mg, dissolved in corn oil, per 20 g mouse body weight, dosed either by IP injection or oral gavage) on 5 consecutive days. iKO (*Kdm2a*^2lox/2lox^; *Actb*^CreERT2/+^) mice were then euthanized at timepoints up to 65 days later. iKO (*Kdm2a*^2lox/2lox^; *CAGG-Cre-ER*^TM/+^) mice were then euthanized at timepoints up to 12 days later. sKO (*Kdm2a*^2lox/2lox^; *Ngn3*^Cre/+^) mice were euthanized 26 days after birth. Tissues were then collected for downstream applications. The following primers were used for genotyping: *Kdm2a* locus: F1 (5′-CCTGGAAATGTGACATAGAAC-3′) and R1 (5′-CTCTCTTCAGCTTCACCTAGAAAC-3′), resulting in products of 741 bp (*Kdm2a*^2lox^ allele), 575 bp (wild-type allele) and 225 bp (*Kdm2a*^1lox^ allele). Mice were housed in Type II long cages containing aspen bedding on IVC racks in rooms with 15-20 air changes per hour and a 12 h light/dark cycle. Temperature was maintained within 20-24 oC with a relative humidity within 45-65%. Food and water was provided ad libitum.

### Histology

Testes and epididymides were fixed in Modified Davidson's solution for 24 h at room temperature and then sectioned at 4 μm and stained with periodic acid-Schiff following routine protocols. Chromogenic immunohistochemistry was performed on testes that were fixed for 24 h at room temperature in 10% neutral buffer formalin (10% NBF; Sigma-Aldrich). Antigens from paraffin-embedded 4 μm sections were retrieved using CC1 (Citrate buffer pH 6.0; Ventana Medical Systems) on the Ventana XT Automated Stainer (Ventana-Roche). Chromogenic staining was then performed on the Ventana XT Automated Stainer. The following antibodies were used: anti-PLZF (Active Motif, #39987 at 20 μg/ml), anti-c-KIT (Cell Signaling, #CST3074 at 0.10 μg/ml), anti-γH2A.X (Cell Signaling, #CST9718 at 0.145 μg/ml), and anti-TRA98 (Abcam, ab82527 at 0.5 μg/ml). Secondary goat anti-rabbit and goat anti-mouse antibodies were purchased from Ventana Medical Systems.

Immunofluorescence was performed using a manual staining protocol. Formalin-fixed-paraffin-embedded sections of 3 or 4 μm testes went through antigen retrieval for 10 or 20 min in antigen retrieval buffer (PMB1-250, SpringBio or 10 mM Sodium Citrate pH 6.0, 0.05% Tween-20) following the manufacturer's protocol. Samples were then blocked in blocking buffer containing PBS, 5% horse serum, 1% BSA and 0.05% Tween-20, and incubated overnight with anti-PLZF (Active Motif, #39987, 1:50), anti-CD117/c-KIT (R&D systems, AF1356-SP, 1:1000), anti-KDM2A (Abcam, ab191387, 1:1000), anti-H3K27me3 (Cell Signaling 9733, 1:1000), anti-H3K36me3 (Cell Signaling 4909, 1:500), anti-H3K36me2 (CosmoBio MABI0332, 1:5000), Lectin PNA Alexa Fluor™ 647 Conjugate (Thermo Fisher, L32460, 1:1000) and anti-SOX9 (Millipore, #AB5535, 1:300) antibodies at 4 °C. Fluorescent secondary antibodies were Alexa 488 or Alexa 647 goat Ig conjugates (Invitrogen). Slides were incubated for 5 min with Hoechst (1:2000 in PBS) or 10 min with DAPI (1:1000 in PBS), mounted (Gold Antifade Mountant, Thermo Fisher #P36935 or Vectashield Antifade Mounting Medium H-1000) and imaged using a Zeiss Confocal microscope or spinning disk confocal scanning unit Yokogawa CSU W1 Dual T2 with 40x/1.3 oil immersion EC Plan Neofluar objective. Representative regions were selected using Fiji software[64].

Tissues were evaluated morphologically by bright-field light microscopy by a veterinary pathologist board-certified by the American College of Veterinary Pathologists (RAV). Histology results were scored in a variety of manners. For morphological findings, tubules were identified by seminiferous tubule stage in periodic acid-Schiff stained testes tissue specimens, with at least 373 individual tubules per mouse scored manually for the presence of degenerative changes, including rounding of cells, cellular hyperchromasia, and/or nuclear pyknosis. For analysis of PLZF and c-KIT positive cells, two cross-sections of testis tissue per mouse were examined, and immunoreactive cells were manually counted in 25 stage VIII seminiferous tubules per cross-section (50 tubules total per mouse). For cleaved caspase-3 analysis, stained tissue sections on glass slides were scanned at 40x magnification. Digital images from two testes sections per animal were analyzed using a color deconvolution algorithm (Aperio ePathology Solutions; Leica Biosystems). A digital pen was used to identify the region of interest and to exclude non-relevant staining (non-specific staining and/or staining of Leydig cells). The relative amount (area) of staining was quantified and expressed as the percentage of strong positive pixels per total stained area of analysis.

Statistical significance was determined using a two-tailed Student's *t*-test unless otherwise stated. Statistical analysis was performed with assistance from GraphPad Prism Software Version 7.0 (GraphPad Software). Statistical significance was given as: (***) *P*-value < 0.001; (**)

*P*-value < 0.01; and (*) *P*-value < 0.05; NS (not significant) *P*-value > 0.05. The results are expressed as the mean value with SD, unless otherwise indicated.

## Immunofluorescence analysis of meiosis samples

We generated drying-down preparations of spermatocytes as described before[34]. For cryo-sections, testes were first fixed overnight at 4 °C in 4% PFA in PBS, then incubated overnight at 4 °C in 30% sucrose and finally embedded in O.C.T. (Sakura Finetek). 10 μm sections were prepared by using a Microm HM355S (Thermo Fisher). For IF staining, both types of preparations were washed with PBS containing 0.1% Tween20 (PBT) and permeabilized (PBS, 0.25% TritonX-100). After several PBS washes, preparations were blocked with PBS, 1% BSA, 0.1% TritonX-100, 5% horse serum and then incubated with primary antibody overnight at 4 °C. Following PBS washes, preparations were incubated for 1 hr with secondary antibodies, washed and counterstained in Vectashield containing 4′,6-diamidino-2-phenylindole (DAPI) (Vector Laboratories). Image acquisition was performed with a Zeiss LSM 700 confocal microscope equipped with 40x and 63×Plan-Apochromat objectives and sequential multitrack scanning using the 405-, 488-, 555- and 639 nm wavelengths of the lasers. We used the following antibodies: rabbit anti-H3K4me3 (1:200 dilution, Millipore 17–614), rabbit anti-H3K9me2 (1:300, Abcam ab1220) rabbit anti-H3K9me3 (1:200, Invitrogen 49-1008), rabbit anti-SCP3 (1:1000, Abcam ab15093), guinea pig anti-HORMAD1 (1:100[65],), mouse anti-γH2A.X (1:5000; Millipore 05636), goat anti-ATR (1:200, Santa Cruz sc-1887).

## RNA extraction, quantitative PCR (qPCR) analyses and cDNA-sequencing

Total RNA was purified using the RNeasy Mini kit (QIAGEN) and reverse-transcribed using a High-Capacity cDNA Reverse Transcription Kit (Thermo Fischer). Quantitative real-time PCR analysis was performed using TaqMan Gene Expression Assays (Applied Biosystems) on a Viia7 real-time thermocycler (Thermo Fischer). Relative mRNA levels were normalized to *beta-Actin* or *Gapdh*. For Sanger sequencing of cDNA, *Kdm2a* locus-specific primers mib06-2a-F (5′- GAGTTTAGC-CACACCAGGCT-3′) and mib06-2a-R (5′- TTGGAACCCGTGTTCGATCT −3′) were used for amplification, products were gel-purified using QIAquick Gel Extraction Kit (QIAGEN), and sequenced using mib05-2a-F (5′- TTAGCCACACCAGGCTTGAG −3′).

## FACS-based isolation of spermatogonia and spermatocytes

Testicular cell suspension was prepared by incubation of seminiferous tubules in 200 U/ml Collagenase type I (Worthington Biochemical LS004196), 5 μg/ml DNAse I (Roche 10104159001), and 0.05%Trypsin (Gibco 25200056) in GBSS (Sigma G9779) as described in ref. 66. Cells were stained with 1/40 anti-CD117/c-KIT PE (eBioscience 12-1171-83), anti-CD324/E-Cadherin eFluor 660 (eBioscience 50-3249-82) and anti-CD49f/Integrin alpha 6 PE-Cyanine7 (eBioscience 25-0459-82) for 1 h at 32 °C with constant shaking and protected from light. After washing, cells were incubated with 20 μg/ml with Hoechst 33342 (Thermo Fischer Scientific H3570) for 1 h at 32 °C with constant shaking and protected from light and then with 30 nM DRAQ7 (Biostatus DR71000) for 5 min on the bench. The stained cells were strained through a 40 μm nylon filter into 5 ml polypropylene tubes and were sorted on a BD FACSAria III cell sorter fitted with a 70 μm nozzle (Becton Dickinson) using a 375 nm laser to excite Hoechst 33342, a 561 nm laser to excite PE and PE-Cy7 and a 633 nm laser to excite DRAQ7 and eFluor600. Cells were first gated for FSC and SSC to exclude debris. Then the live cells were selected based on the absence of DRAQ7 signal detected using 755LP, 780/60BP. Then we gated for cells positive for Hoechst 33342 emission, which was detected using a 670LP (Hoechst-Red) and 450/20 BP (Hoechst-Blue). We gated for the Hoechst-Red[low]

and Hoechst-Blue[mid] population, which is enriched for 2 N spermatogonia. Fluorescence for PE was detected using a 582/15BP filter, for PE-Cy7 was detected using 735LP,780/60BP filter and for eFluor660 using a 660/20BP filter. Undifferentiated spermatogonia were sorted as CD324[high], CD49f[high], CD117[low] and differentiated spermatogonia were sorted as CD324[low], CD49f[low], CD117[high]. We gated for the Hoechst-Red[mid] and Hoechst-Blue[high] population, which is enriched for Leptotene and Zygotene spermatocytes[67].

Cells (5,000 to 10,000) were sorted directly into either cell lysis buffer supplemented with 1% beta-mercaptoethanol (Norgen Single Cell RNA Purification Kit, cat. No. 51800, Norgen Biotek Corp.) for transcriptomic analysis or nuclear isolation buffer (Sigma-Aldrich NUC101) for genomic analysis in Eppendorf® LoBind microcentrifuge tubes.

## RNA-seq library generation

Total RNA was isolated from sorted cells using the Norgen Single Cell RNA Purification Kit. Per sample, we used RNA equivalent to 10,000 cells to generate RNA-seq libraries using the Ovation® SoLo RNA-Seq Library Preparation Kit (Nugen, 0501-32) following the manufacturer's instructions. Libraries were sequenced in 2 × 38 bp paired-end mode with NextSeq 500 sequencing technology (Illumina), per the manufacturer's recommendations.

## Ultra-low input native chromatin immunoprecipitation (ULI-NChIP)

To profile histone modifications in spermatogonial cells, we followed the previously published ultra-low input native ChIP-seq (ULI-NChIP-seq) protocol[22] with minor modifications. Briefly, 5000–7500 spermatogonia were FACS-sorted directly in 20 μl nuclear isolation buffer (Sigma-Aldrich NUC101) in Eppendorf® LoBind microcentrifuge tubes. The nuclei were centrifuged at 500 × min at 4 °C in a swing bucket centrifuge, and the volume of the sample was reduced to 10 μl. The sample was snap frozen in liquid nitrogen and stored at – 80 °C freezer until further processing. The sample was thawed on ice, and 1 μl of TS buffer (5% Triton X-100, 5% Sodium deoxycholate) was added for 10 minutes to ensure permeabilization of nuclei. 40 μl of MNAse digestion buffer (3.33 gelUnits/μl Micrococcal Nuclease (NEB M0247S), 1.25X Micrococcal Nuclease buffer, 6.25% PEG 6000, 0.85 mM DTT) was added to the sample for 10 minutes at 25 °C to fragment the chromatin. Reaction was stopped by the addition of 5 μl 100 μM EDTA, and the sample was placed on ice. The sample was diluted by the addition of 145 μl ChIP buffer (20 mM Tris-HCl pH 8.0, 2 mM EDTA, 150 mM NaCl, 0.1% Triton X-100, 1X EDTA-free protease inhibitor cocktail). To ensure release of chromatin fragments, the sample was sonicated using the Diagenode Bioruptor Plus® for 3 cycles of 5 seconds ON and 30 seconds OFF at low output. Then, chromatin was pre-cleared with 10 μl of 1:1 protein A:protein G Dynabeads (Thermo Fisher Scientific, 10001D and 10003D). The precleared chromatin sample was incubated with 1 μg anti-H3K36me2 (Upstate 07-369), 0.5 μg of anti-H3K4me3 (Millipore 17–614) or 0.75 μg of anti-H3K27me3 (Cell Signaling 9733) conjugated with 10 μl of 1:1 protein A: protein G Dynabeads overnight at 4 °C on a rotator. Chromatin-Antibody-Dynabeads complexes were washed once with 500 μl of ChIP buffer, 3 times with 1 ml of Low Salt wash buffer (20 mM Tris-HCl (pH 8.0), 0.05% SDS, 1% Triton X-100, 2 mM EDTA and 150 mM NaCl) and 3 times with 1 ml High Salt buffer (20 mM Tris-HCl (pH 8.0), 0.05% SDS, 1% Triton X-100, 2 mM EDTA and 250 mM NaCl). Chromatin was eluted in 40 μl of Elution buffer (100 mM NaHCO3 and 1% SDS) for 90 min at 65 °C. DNA from eluted material was purified by 2X volumes of Ampure XP DNA purification beads (Beckman Colter, A63881) and resuspended in 25 μl nuclease-free H2O. Sequencing was performed on an Illumina HiSeq 2500 machine with single-end 50-bp read length (Illumina), per the manufacturer's recommendations.

### Reduced representation bisulfite sequencing (RRBS)

DNA was extracted from 10.000 spermatogonia cells by QIAamp DNA Micro Kit (Qiagen # 56304) according to the manufacturer's instructions. We prepared RRBS libraries using 10–20 ng of DNA and following the previously published method[68]. Briefly, DNA was digested with 0.9 µl MspI Fast Digest (Thermo Scientific FD0544) in 1X Tango buffer in a volume of 18 µl for 6 h at 37 °C. DNA ends were repaired by adding 1 µl Klenow Fragment, exo− (Thermo Scientific EP0421) and 1 µl of nucleotide end-repair mix (1 mM dATP; 0.1 mM dGTP; 0.1 mM dCTP prepared in 2X Tango buffer) and incubated for 45 min at 37 °C, followed by enzyme heat inactivation at 75 °C for 15 min. Then, 6.25 nM of 5mC sequencing adapter (NEB E7535S) was ligated with 1 µl T4 DNA Ligase, HC (Thermo Scientific EL0013) and 0.5 µl ATP solution (Thermo Scientific R0441) for 16 h at room temperature. The looped adapter ligated DNA fragments were subjected to USER (NEB M5508) digestion at 37 °C for 20 min and used directly for bisulfite treatment using the Imprint DNA Modification Kit (Sigma MOD50) according to the manufacturer's 2-step modification procedure. The modified DNA was amplified with 13 cycles of PCR using Kapa U+ polymerase (Roche 07959052001) and indexed primers (NEB) and purified with 1.8X Ampure XP beads. Libraries were sequenced on a HiSeq 2500 sequencing platform, generating 50 bp single-end reads, per the manufacturer's recommendations.

### Computational analyses

Computational analyses were performed using publicly available Bioconductor software run in RStudio (R version 4.2.1).

*Reference genome and annotations*: The UCSC genome assembly GRCm38/mm10 was used as a reference. Transcript annotations were based on the UCSC knownGene database and were obtained from the Bioconductor package TxDb.Mmusculus.UCSC.mm10.knownGene (version 3.2.2). Genomic coordinates of CpG islands were obtained from the UCSC genome annotation database at http://hgdownload.soe.ucsc.edu/goldenPath/mm10/database/ (file name cpgIslandExt.txt.gz).

*Sequence Data Alignment and Processing*: ULI-NChIP-seq reads were first processed using TrimGalore (version 0.6.2) to trim adapter and low-quality reads with settings (--stringency 3). Trimmed reads were then aligned to the mouse genome build mm10 using STAR (version 2.5.0a) with settings (--alignIntronMin 1 --alignIntronMax 1 --alignEndsType EndToEnd --alignMatesGapMax 1000 --outFilterMatchNminOverLread 0.85). Aligned reads were deduplicated using SAMtools (version 1.10) with standard settings. Uniquely mapped reads were counted on genomic regions [gene transcriptional start sites (TSS) +/− 1 kb] using the function QuasR::qCount (mapqMin = 255 L)(version 1.36.0). $\log_2$RPKM values were calculated for each TSS and normalized between biological samples of *Kdm2a* iKOs and Ctrl using the function limma:normalizeBetweenArrays (method = " cyclicloess",cyclic.method = "fast") (version 3.52.2). Normalized $\log_2$RPKM values were transformed back to read counts and were processed by the edgeR package (version 4.2.1)[69] to identify differential enrichment of histone modification on TSS among experimental groups and calculate $\log_2$ fold changes with standard parameters. Statistical significance cutoffs for a TSS to be considered as differentially enriched were set as follows: $\log_2$FC > |1| and FDR < 0.05. QuasR::qProfile (binSize = 50) was used to obtain ULI-NChIP-seq read profiles for TSS +/− 5 kb.

Nugen Ovation Solo RNA-seq reads were first processed using TrimGalore (version 0.6.2) to trim adapters and low-quality reads with settings (--clip_R1 3, --stringency 3). Trimmed reads were then aligned to the mouse genome build mm10 using STAR (version 2.5.0a) with settings (--outFilterMismatchNmax 6). Duplicated reads were marked/removed using nudup.py (version 2.2) with settings (-s 8, -l 8). The QuasR::qCount function was used for quantification of uniquely mapped reads on genes, and the 'junction' reportLevel argument was specified for quantification of exon-exon junction reads. The resulting read count data were processed by edgeR to identify differentially expressed genes among experimental groups. Genes with CPM < 1 were excluded from the analysis. Genes with $\log_2$FC > |1| and FDR < 0.05 were considered to be significantly changed. For principal component analysis in Supplementary Fig. 6a the package PCAtools (version 2.8.0) was used.

RRBS reads were first processed using TrimGalore (version 0.6.2) to trim adapters and low-quality reads with settings (--rrbs, --stringency 3). Trimmed reads were then aligned to the mouse genome build mm10 using the qAlign function from the QuasR package. Methylated and unmethylated read counts for each CpG were extracted from the BAM files using the function qMeth from the QuasR package. CpGs covered with less than 5 reads were discarded from downstream analysis. In addition, TSS +/− 1 kb genomic windows with less than 5 "covered CpGs were discarded from downstream analysis. The summed read count of methylated and unmethylated CpGs was calculated for each retained TSS +/− 1 kb genomic window. The resulting read count data were processed by edgeR to identify differentially methylated tiles among experimental groups. Tiles with methylation percentage change > |20%| and false discovery rate-adjusted FDR < 0.05 were considered as significantly changed. The percentage of methylation was calculated as the fraction of methylated CpG read counts to the total CpG read counts for a TSS +/− 1 kb genomic window.

Clustering of genes in Fig. 5a was performed with hclust and dist functions from the stats package (version 4.2.1) using the "Ward.D" method and "Euclidean" distance. As an input matrix, we used the z-score of RNAseq $\log_2$RPKM gene expression in *Kdm2a* iKO and Ctrl SgD, the z-score of ULI-NChIPseq $\log_2$RPKM enrichments at gene TSS +/− 1 kb in *Kdm2a* iKO and Ctrl SgD and the z-score of RRBS DNA methylation percentage at gene TSS +/− 1 kb in *Kdm2a* iKO and Ctrl SgD.

Gene ontology analysis was performed using the Mouse GO slim subset and the package topGO (version 2.48.0).

Heatmaps were generated in R using the ComplexHeatmap[70] package. Other plots were generated using the ggplot2 and GGally packages. Bigwig files for all sequencing experiments were generated using the qExportWig function from the QuasR package. Genome browser tracks were generated using the UCSC genome browser (http://genome.ucsc.edu).

Single cell gene expression was visualized by loading cloupe files from[14,71] onto the Loupe Browser (v8.1.2, 10x Genomics).

### Reporting summary

Further information on research design is available in the Nature Portfolio Reporting Summary linked to this article.

## Data availability

The datasets produced in this study are available in the following database: Gene Expression Omnibus GSE214682. Source data files, including the relevant raw data from each graph, are available at the general repository Figshare (https://doi.org/10.6084/m9.figshare.29256725). In addition, we provide a Source Data file with transcriptomic data of genes and chromatin data of their corresponding TSS +/− 1 kb, which are examined in the current work. The raw FACS generated in this study have been deposited at the general repository Figshare (https://doi.org/10.6084/m9.figshare.29269619). The raw image data files generated in this study have been deposited in the Biostudies database under accession code "S-BSST2090" (https://www.ebi.ac.uk/biostudies/studies/S-BSST2090). Source data are provided in this paper.

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

## Acknowledgements

We thank Birgitte Lucas (FMI) for the preparation and sequencing of RNA sequencing libraries, Laurent Gelman (FMI) for microscopy support, and Fred Zilbermann (FMI) and FMI animal facility members for excellent support in handling experimental animals. We thank Kelly Blanchard, John Halupowski, Vanessa Davis (Novartis lab animal services), Alyssa Riley (Novartis study support), Jolanta Dubauskaite, Gerlinde Wussler and Paul Gedman for critical support with mouse studies, Mei Li, Qing Fang and Hong Lei for supporting the mouse model generation, and Peter K. Nicholls (MIT/ Whitehead Institute) for advice and critical discussion of the manuscript. We thank Filippo Rijli (FMI) for providing the *CAGG-Cre-ER*$^{TM}$ mouse line and Attila Toth (Technische Universität Dresden, Germany) for providing the anti-HORMAD1 antibody. M.T.B. thanks the NIBR postdoctoral program for support. H.R. thanks the postdoctoral fellowships from the Peter und Traudl Engelhorn program and the Marie-Curie FP7 (call identifier FP7-PEOPLE-2013-IEF). S.C. thanks funding from the JSPS "Strategic Young Researcher Overseas Visits Program for Accelerating Brain Circulation" (Number S2503). A.H.F.M.P. gratefully acknowledges funding from the Novartis Research Foundation and the Swiss National Science Foundation (31003A_146293, 31003A_172873).

## Author contributions

M.T.B., A.H.F.M.P., and T.B.N. conceived, designed and directed the study with critical support of G.F., X.M., M.A.L., and J.T. M.T.B. performed molecular and cell biology and in vivo experiments. G.F. performed transcriptomic and epigenomic experiments. G.F., A.R., and H.R. performed computational analyses with advice of M.B.S. and A.H.F.M.P. S.C. characterized the meiotic phenotype. C.-Y.L. and G.F. developed a spermatogonial and meiotic FACS sorting protocol with the support of P.A.K. and H.K. P.A.K. and M.T.B. performed immunofluorescence staining and imaging. K.W. generated (immuno-)histological preparations and evaluated histological preparations with advice by P.C., M.T.B., R.A.V., D.G.d.R., and A.H.F.M.P. T.C. and T.B.N. generated conditional *Kdm2a*-deficient mice. H.R., A.R., S.C., and C.-Y.L. contributed equally to this work. M.T.B., G.F., T.B.N., and A.H.F.M.P. wrote the manuscript with input of other authors.

## Competing interests

M.T.B., K.W., T.C., X.M., M.A.L., R.A.V., P.C., J.T., and T.B.N. are current or past employees of Novartis. The remaining authors declare no competing interests.

## Additional information

**Michael T. Bocker**[1,9], **Grigorios Fanourgakis** [2,9], **Kristie Wetzel**[3], **Pavel A. Komarov** [2,4], **Hélène Royo**[2,5], **Alexia Rohmer**[2], **Sunwoo Chun**[2,4,6], **Ching-Yeu Liang**[2,4], **Hubertus Kohler**[2], **Taiping Chen** [1,7], **Xiaohong Mao**[3], **Mark A. Labow**[1], **Reginald A. Valdez**[1], **Michael B. Stadler** [2,5], **Dirk G. de Rooij**[8], **Paola Capodieci**[3], **John Tallarico** [1], **Antoine H. F. M. Peters** [2,4,10] ✉ & **Thomas B. Nicholson** [3,10] ✉

[1]Discovery Sciences, Novartis Biomedical Research, Cambridge, MA, USA. [2]Friedrich Miescher Institute for Biomedical Research (FMI), Basel, Switzerland. [3]Disease Area X (DAx), Novartis Biomedical Research, Cambridge, MA, USA. [4]Faculty of Sciences, University of Basel, Basel, Switzerland. [5]Swiss Institute of Bioinformatics, Basel, Switzerland. [6]Graduate School of Agricultural and Life Sciences, The University of Tokyo, Tokyo, Japan. [7]Department of Epigenetics and Molecular Carcinogenesis, The University of Texas MD Anderson Cancer Center, Smithville, Texas, USA. [8]Reproductive Biology Group, Division of Developmental Biology, Department of Biology, Faculty of Science, Utrecht University, Utrecht, the Netherlands. [9]These authors contributed equally: Michael T. Bocker, Grigorios Fanourgakis. [10]These authors jointly supervised this work: Antoine H. F. M. Peters, Thomas B. Nicholson. ✉e-mail: antoine.peters@fmi.ch; thomas.nicholson@novartis.com

