## [Peer Review file · Nature Communications]

The histone H3 lysine 36 demethylase KDM2A/FBXL11 controls Polycomb-mediated gene repression and germ cell development in male mice

Corresponding Author: Professor Antoine Peters

Version 0:

Reviewer comments:

Reviewer #1

(Remarks to the Author)

I found this manuscript to be concise and very clearly written. The study design was sufficient and I would not suggest further studies. The figures were clear and compelling. In short, I found this to be an interesting and useful study in further understanding germ cell biology and spermatogenesis.

Reviewer #2

(Remarks to the Author)

The manuscript by Bocker et al., describes the phenotypic and molecular characterization of conditional mouse knockout of the histone H3 lysine 36 demethylase KDM2A. The authors convincingly demonstrate that the knockout has a defect during spermatogenesis and suggest that this is a consequence of inappropriate expression of a class of PRC regulated genes. In general, this is a well done study with insight into the unique gene expression patterns required for spermatogenesis.

The authors should consider the following:

1. It would be beneficial to the non-mouse investigator to refer to the timeline/stages of spermatogenesis (Fig 2g) in the introduction (page 2; line 61)
2. The authors report there are defects in chromosome synapsis and meiotic recombination; however, they never directly address whether the defects they observe are a consequence of failure in MSCI. There images in figure 5 (and related supplementary figures) are certainly consistent with this. At a minimum this needs to be included in the discussion. Further, the authors could explain why it would be difficult to directly address the role of KDM2A in oogenesis.
3. Did the authors look for enrichment in X chromosome expressed genes in their RNA seq data from the conditional knockout? If there is a failure in MSCI, this should result in upregulation of X linked genes known to be deleterious to spermatogenesis. It would be of interest to specifically look at this.
4. I would also like to see more of the discussion of the role of H3K36me2/me1 in transcription activation/repression/DNA integrity.

Reviewer #3

(Remarks to the Author)

In this manuscript, the authors aimed to demonstrate the essential role for the histone H3 lysine demethylase KDM2A/FBXL11 in spermatogenesis during development in both the tamoxifen inducible and the constitutive Ngn3-Cre driven knockout mouse models. The authors used multiple experimental approaches, including histological and molecular methods, characterized the spermatogonial phenotypes in great details, and well-documented the spermatogonial defects in those conditional knockout mice. They present a series of interesting new findings and propose a potentially important model of this biologically important event. The manuscript is well written and the data included are very clear, there are several serious concerns that need to be addressed before publication. Specific points are listed below.

(1) KDM2A has been known to play an essential role in mouse embryonic development by regulating cell proliferation and survival (Kawakami et al, 2015). Therefore, it is not so surprising to know that Kdm2a deficiency in mice causes defects in spermatogonial cell proliferation and differentiation, delayed cell cycle progression and apoptosis. What remains largely unknown is the underlying molecular mechanisms. It appears that KDM2A plays an important role in controlling expression of numerous genes that are responsible for spermatogonial differentiation and progression through meiotic prophase. However, the study only scratched the surface with little mechanistic insight.

(2) KDM2A is a histone demethylase specific for mono- and di-methylated histone H3 lysine 36. However, in this manuscript the authors did not examine changes in this histone mark in spermatogonial cells at different developmental stages upon loss of Kdm2a.

(3) The authors performed RNA-sequencing assays using RNA from purified spermatogonia and spermatocytes, and identified groups of candidates. Unfortunately, the data included in the paper are largely descriptive with no validation of selected candidates. For example, while kdm2a is primarily required for gene repression, the authors listed one group of genes in Figure 6 that failed to be induced in iKO cells, but did not perform any further validation of expression of these genes, and simply mention "their delayed or impaired activation might underlie the impairment of meiotic progression and cell death observed for Kdm2a iKO spermatocytes". It would be interesting and very important to know whether differential gene expression is correlated with local changes in the level of methylated H3K36 in Kdm2a deficient cells.

(4) It has been shown that KDM2A is ubiquitously expressed during embryogenesis and in several adult tissues (Fukuda et al, 2011; Kawakami et al, 2015). Whether or not the same is true for KDM2A in spermatogonial tissue remains unclear. It would be very helpful if the authors could show the expression pattern of endogenous KDM2A during proliferation and differentiation of spermatogonia.

(5) The authors provide clear in vivo evidence showing that KDM2A is indispensable for the generation and maintenance of differentiating spermatogonia. Given that the culture of spermatogonial stem cells of mice was proved to be successful (Kanatsu-Shinohara M, 2007), it may be very useful if the authors could confirm the in vivo data by ex vivo culture of the spermatogonial stem cells from control and Kdm2a deficient mice.

(6) In page 4 line 106, the authors need to mention which Cre driver was used to generate the homozygous kdm2a^{1flox/1flox} mice that died at mid-gestation.

(7) In Fig 1d, it would be better to replace the absolute weight with normalized weight (normalized by body weight).

Version 1:

Reviewer comments:

Reviewer #2

(Remarks to the Author)

The revised manuscript by Bocker et al., describes the phenotypic and molecular characterization of conditional mouse knockout of the histone H3 lysine 36 demethylase KDM2A. The authors convincingly demonstrate that the knockout has a severe defect during spermatogenesis and suggest that this is a consequence of inappropriate expression of a class of PRC regulated genes. The authors have done an excellent job addressing the previous reviews, including new experimental data and in the presentation. This is an important study with implications for understanding how spermatogenesis is regulated at the transcriptional level and will be of interest to the broad audience of Nature Communications.

Reviewer #4

(Remarks to the Author)

The manuscript entitled "The histone H3 lysine 36 demethylase KDM2A/FBXL11 controls PRC2-mediated gene repression and germ cell development in male mice" by Michael et al. exploited both germline-specific and tamoxifen-induced mouse models to explore the roles of Kdm2a, a demethylase for H3K36me_{2/1}, in the germline development in mouse testicular development. They found that Kdm2a serves an important role during the differentiation of spermatogonia. Mechanistically, Kdm2a represses the PRC2-target gene expression to safeguard the correct spermatogonia-to-meiotic transition. This study is overall well-designed, the results are clearly described with sufficient mechanistic evidence. The data strongly support the major conclusions. I believe it is a significant study in the male reproductive field. I have only minor comments as below:

1. With the easy access to the available single-cell RNA-seq datasets, please also include a plot showing the dynamic expression levels of KDM2a mRNA throughout germline development. It would be interesting to see whether Kdm2a is much higher in differentiating spermatogonia since it is functional at this stage.
2. Could the authors explain why the c-kit signal was so strong in the interstitial cells in the testes in the immunofluorescence staining images, e.g. Figure 1F, Supple Fig.2 ?
3. For calculation of exon expression levels in supple Fig.6B. the authors mentioned that there was a 35% reduction in the total read numbers in the iKO germ cells. Were these calculated against averaged exon levels ? we usually observed the

compensatory increase in the levels of the adjacent exons around the floxed exons in the conditional KO models from the RNA-seq data. This could be checked by showing a snapshot of the IGV track of the RNA-seq data.

4. In Supple Fig.9F, have you ever compared the H3K36me2 peak intensity between Ctrl and iKO in 10 day SgD ? It appears that the peak intensity levels are comparable across the gene bodies when comparing Ctrl with iKO. A zoom-in view of IGV track in the promoter regions would be more informative to the readers. In addition, have the authors exploited the CUT&Tag to do the H3K36me2 modification? It has been reported to be powerful to identify the promoter region alongside the gene body localization (<https://doi.org/10.1038/s41467-022-30940-1>)

Reviewer #5

(Remarks to the Author)

In this study, the authors describe a critical role for the histone demethylase KDM2A in supporting adult spermatogonial differentiation and meiotic progression. Mechanistically, they performed multiple omics analyses on isolated spermatogonia and spermatocytes and demonstrated that KDM2A is required for PRC2-mediated repression by increasing H3K36me2 and decreasing H3K27me3 promoter occupancy. Furthermore, they showed that Kdm2a ablation in spermatocytes disrupts meiotic progression, accompanied by impaired chromosome synapsis completion and DSB processing, likely due to altered chromatin states and impaired X-linked gene repression. However, the main criticism of the study is its novelty, as a similar study has already revealed the function of KDM2A in meiosis (PMID: 39160277).

Other major points:

1. For Figure 1F, the cells labelled by the c-KIT antibody are probably not spermatogonia, but rather interstitial cells of the testis. Please specify the cell type of the KDM2A positive cells.

2. It is better to determine whether KDM2A is expressed in testicular cells (e.g. Sertoli cells and Leydig cells) in addition to germ cells. Although the authors found that the Actb-CreERT2-induced Kdm2a knockout mice had a similar phenotype to the germ cell-specific Kdm2a knockout mice driven by Ngn3-cre, it cannot be ruled out that KDM2A may play a role in other testicular cells for spermatogenesis.

3. Did abnormal meiosis occur in the Kdm2a sKO mice during the first wave of spermatogenesis, similar to that observed in Kdm2a iKO mice?

4. It is still unclear whether KDM2A ablation affects the overall changes in histone modifications in germ cells, so it is necessary to perform IF analysis in testicular sections using H3K36me1/2, H3K4me3 and H3K27me3 antibodies.

5. The authors identified 779 up-regulated genes associated with "morphogenesis" and "cell adhesion" in the KDM2A-ablated differentiated spermatogonia, explaining the abnormal spermatogonial differentiation phenotype. Are the "cell death" related genes such as Tnf, Egfr and Myc specifically upregulated in the differentiated spermatogonia of Kdm2a iKO mice? Are their protein expression levels altered?

6. For Figs. 5A, 5B, how many genes in cl.2 to 4 and cl.7 are included in the upregulated gene sets shown in Fig. 4E? Since the authors conclude that KDM2A is required for PRC2-mediated repression, why did only cl.3 genes show reduced H3K27me3 abundance?

7. Given that some of the differentially expressed genes (DEGs) in Figure 4E encode important transcription factors (Pou5f1, Wt1, Zic2, Eomes, Lef1 and Twist1), I don't think that all DEGs are directly bound and regulated by KDM2A. It is better to clarify which DEGs are directly bound by KDM2A and analyse the changes in histone modification (e.g. H3K36me2, H3K4me3, H3K27me3) on these KDM2A-bound genes.

Minor comments:

1. Why does the Wt1 gene, which encodes a Sertoli cell marker protein, appear in the set of genes that are up-regulated in KDM2A-ablated differentiated spermatogonia?

2. Line 944, the legend for Fig 4F should be corrected to "iKO ScLZ versus Ctrl ScLZ..."

Reviewer #6

(Remarks to the Author)

The authors have addressed Reviewer #3's comments comprehensively, particularly in providing mechanistic insights into the role of KDM2A in spermatogonial differentiation and meiotic progression. Below are my observations and recommendations:

1. Mechanistic Insights:

The rebuttal effectively clarifies how KDM2A-specific effects, particularly its role in silencing PRC2 target genes, are pronounced in spermatogonial cells despite KDM2A's broader biological roles. The inclusion of ChIP-seq data on

H3K36me2 and analyses of CGI-rich promoter dynamics strengthens the mechanistic narrative.

2. Histone Modification Dynamics:

The added ChIP-seq data on H3K36me2 addresses the reviewer's concerns, providing robust evidence for promoter-level changes in differentiating spermatogonia. The insights into changes in other histone marks (e.g., H3K4me3 and H3K27me3) also enrich the mechanistic understanding. Still, a deeper discussion of how these findings align with existing epigenetic paradigms in spermatogenesis would be beneficial.

3. Expression Pattern of KDM2A:

The addition of immunofluorescence data elucidating KDM2A expression during different spermatogenic stages is commendable. This visual evidence helps link KDM2A's presence with its functional role in differentiation and meiotic processes.

4. In vitro Validation via Stem Cell Culture:

While the authors' rationale for not pursuing in vitro experiments is reasonable, a more detailed explanation of limitations (e.g., challenges in synchronizing differentiation stages in culture) would add clarity.

Overall, the authors have addressed major concerns raised by Reviewer #3 thoughtfully and with substantial new data. I support to accept the revised manuscript.

Version 2:

Reviewer comments:

Reviewer #4

(Remarks to the Author)

My questions have all been adequately addressed! It is now suitable for acceptance.

Reviewer #5

(Remarks to the Author)

The authors have comprehensively addressed the comments and discussed the differences between the current study and other previous publications. Despite the fact that the originality of the work has been compromised to some extent, it is nevertheless deserving of publication within this field.

Reviewers' comments:

Reviewer #1 (Remarks to the Author):

I found this manuscript to be concise and very clearly written. The study design was sufficient and I would not suggest further studies. The figures were clear and compelling. In short, I found this to be an interesting and useful study in further understanding germ cell biology and spermatogenesis.

Thank you very much for your positive and encouraging feedback on our manuscript. We are pleased to hear that you found the study design sufficient, the figures clear and compelling, and the manuscript well-written. Your acknowledgment of the study's contribution to the understanding of germ cell biology and spermatogenesis is greatly appreciated. Nevertheless, we have extensively revised our manuscript based on the comments and suggestions of the other reviewers which we think has further strengthened our paper.

Reviewer #2 (Remarks to the Author):

The manuscript by Bocker et al., describes the phenotypic and molecular characterization of conditional mouse knockout of the histone H3 lysine 36 demethylase KDM2A. The authors convincingly demonstrate that the knockout has a defect during spermatogenesis and suggest that this is a consequence of inappropriate expression of a class of PRC regulated genes. In general, this is a well done study with insight into the unique gene expression patterns required for spermatogenesis.

The authors should consider the following:

1. It would be beneficial to the non-mouse investigator to refer to the timeline/stages of spermatogenesis (Fig 2g) in the introduction (page 2; line 61)

Thank you for your suggestion. To enhance the clarity and accessibility of our manuscript, particularly for readers who may not be as familiar with mouse spermatogenesis, we have added a reference to the timeline and stages of spermatogenesis in the Introduction section referring to the relevant schemes in Figures 1e and 2g.

Page 3 line 66-77 have been revised to:

“Seminiferous tubules can be separated into 12 recurring stages (conventionally noted by Latin numerals I-XII), based on the composition of developing germ cells (Fig. 1g) (2). Type As spermatogonia give rise to proliferating type A paired (Apr) and type A aligned spermatogonia (Aal-4, Aal-8, Aal-16 or Aal-32), which remain interconnected by cytoplasmic bridges and that can still self-renew. Type As, Apr and Aal spermatogonia can be found in any stage of the seminiferous tubules. At stage VIII, type Aal spermatogonia form differentiating spermatogonia (subsequently type A1, A2, A3, A4, intermediate (In) and type B spermatogonia) (3). Type A1-4 spermatogonia are found in stages IX to II, and In spermatogonia are found in stages II-III which differentiate into B spermatogonia in stage IV and V (Fig. 2g). Following six mitotic spermatogonial divisions, B spermatogonia form preleptotene spermatocytes at stage VII which enter meiosis at stage VIII to form leptotene spermatocytes.”

2. The authors report there are defects in chromosome synapsis and meiotic recombination; however, they never directly address whether the defects they observe are a consequence of failure in MSCI. There images in figure 5 (and related supplementary figures) are certainly consistent with this. At a minimum this needs to be included in the discussion. Further, the authors could explain why it would be difficult to directly address the role of KDM2A in oogenesis.

Thank you for your suggestion to discuss further the implication of meiotic silencing of unsynapsed

chromosomes (MSUC) and meiotic sex chromosome inactivation (MSCI) in our model. We have accordingly updated the Discussion page 19-20 line 499-544:

“Finally, we measured notable changes in X-linked gene expression in Ctrl and *Kdm2a* iKO spermatocytes as well as a developmental arrest of *Kdm2a* iKO spermatocytes occurring at the pachytene stage of meiotic prophase I. Previous cytological research on meiotic prophase substages identified significant lower levels of chromatin-associated RNA Polymerase II in leptotene and zygotene compared to pachytene and diplotene spermatocytes (45). In line, an extensive RNA-seq study reported a transient global reduction in gene expression in preleptotene and leptotene spermatocytes compared to preceding spermatogonial and later stage spermatocyte populations (46). We hypothesize that such transient genome-wide transcriptional downregulation may result from BRCA1 and ATM signaling at DSBs and unsynapsed chromosome axes taking place in very early meiotic cells (46,47). Subsequently, as DSB repair and synapsis of autosomal chromosomes proceeds from zygotene to pachytene stages, autosomal genes become reactivated, while X-linked genes become further repressed at pachynema and diplonema in a process known as meiotic sex chromosome inactivation (MSCI) (27,45,48). In our study, the FACS sorted Ctrl and *Kdm2a* iKO ScLZ populations contain a mixture of early meiotic prophase I spermatocytes (10% preleptotene, 40% leptotene, 40% zygotene and 10% early pachytene spermatocytes, data not shown). In Ctrl ScLZ cells, we measured a reduction in the expression ratio of chromosome X-linked (X) genes versus autosomal (A) genes, as compared to spermatogonia (Fig. 6d). Following our hypothesis raised above, we propose that ensuing the transient global repression in leptotene spermatocytes, the autosomal genes in zygotene and early pachytene cells of the Ctrl ScLZ population became upregulated while X-linked expression did not, thus explaining the observed reduction in the X:A gene expression ratio.

Failure to complete autosomal chromosome synapsis by the end of the zygotene stage and retention of the DNA damage response factors at unsynapsed autosomes are thought to drive meiotic silencing of unsynapsed chromatin (MSUC) which, depending on its extent, may prevent subsequent MSCI (47-49). While meiotic DSB formation and chromosome synapsis were initiated in *Kdm2a*-deficient spermatocytes, these processes failed to be completed at the developmental timing of the pachytene stage, resulting in persisting unsynapsed chromosome axes labeled by the synapsis surveillance protein HORMAD1, the ATR DNA damage kinase, and chromatin highly labeled with γ H2A.X (Fig. 6a, 6b, 6c). Although MSCI is thought to begin in pachytene spermatocytes, we detected an upregulation of X-linked genes already in *Kdm2a* iKO ScLZ cells (Fig. 6d). Accordingly, the pachytene-like spermatocytes in *Kdm2a*-mutant mice failed to compartmentalize the X and Y chromosomes into a peripheral nuclear subdomain called the sex-body (Fig. 6b, 6c), further supporting that the X and Y chromosomes were not properly silenced. Mice bearing mutations in proteins associated with DNA damage response such as BRCA1 (50), MDC1 (51) and H2A.X (52) also exhibit aberrant meiotic chromosome synapsis, MSCI failure and developmental arrest around mid-pachytene stage, as we observed for *Kdm2a*-mutant spermatocytes. In human (53) and yeast (54) cells, the di-methylation of H3K36 by SETMAR or MMSET at DSBs or telomeres, respectively, recruits Ku70 and NBS to promote non-homologous end joining (NHEJ). We hypothesize that during meiosis, KDM2A may be required to modulate H3K36me2 levels at DSBs to promote DNA damage repair via homologous recombination. Collectively, the persistence of meiotic pairing defects, the failure to complete meiotic DSB repair and to induce MSCI, possibly in conjunction with an incomplete maturation of meiotic chromatin, as exemplified by persistent elevated levels of H3K4me3 and H3K9me2, likely underlie the meiotic developmental arrest of *Kdm2a*-deficient pachytene-like spermatocytes.”

In addition, we refer to 2 recent studies by Kawamura Y.K., et al. (2024) from our lab and by Xiong X., et al (2022) from Southwest Minzu University, examining the role of KDM2A in oogenesis. Both studies agree that the oocyte development is not affected by conditional depletion of *Kdm2a* using *Zp3-Cre* in the female germline. Introduction page 4 line 97-100:

“In growing mouse oocytes, KDM2A and KDM2B prevent DNMT3A-catalyzed de novo DNA methylation and recruit variant Polycomb Repressive Complex 1 (vPRC1) at CGI-gene promoters, thereby controlling H2A mono-ubiquitin deposition, vPRC1-dependent gene repression as well as downstream-acting PRC2-mediated H3K27me3 deposition (10).”

And discussion page 17 line 437-439:

“In contrast, the development of *Kdm2a*-deficient oocytes is not grossly affected (10,33), illustrating a major sexual dimorphism in the function of KDM2A in both germ lines.”

3. Did the authors look for enrichment in X chromosome expressed genes in their RNA seq data from the conditional knockout? If there is a failure in MSC1, this should result in upregulation of X linked genes known to be deleterious to spermatogenesis. It would be of interest to specifically look at this.

Thank you for your suggestion to harness our transcriptome data to identify potential MSC1 failure. We undertook 2 different analytical approaches. First, we estimated the ratio of the median expression of X-linked genes versus median expression of autosomal genes in all cell types and genotypes as a proxy of sex chromosome activity. Secondly, we assessed the expression and the promoter chromatin state of each X-linked gene. These data are presented in Figure 6d and Supplemental Figure 11. We have revised the Result section page 15-16 line 396-414 accordingly:

“We then compared the median expression ratio of X-linked versus autosomal genes, which allows to track sex chromosome activity (Fig. 6d). We noticed that the X:A ratio in Ctrl ScLZ was decreased compared to the Ctrl spermatogonia (Fig. 6d). The downregulation of X-linked expression in Ctrl ScLZ cell population relative to SgD is notable since meiotic sex chromosome inactivation (MSCI) is thought to initiate silencing of X-linked gene expression in pachytene stage spermatocytes (29). In contrast, we observed that the X:A ratio in *Kdm2a* iKO ScLZ samples remained similarly high as observed in the preceding *Kdm2a* iKO spermatogonia (Fig. 6d). Accordingly, we detected higher expression of X-linked genes in *Kdm2a* iKO versus Ctrl ScLZ (Supplemental Fig.11a, 11b). The failure to induce down-regulation of X-linked genes in *Kdm2a* iKO ScLZ population may point to a role of KDM2A in DNA damage response induced gene silencing. This role is distinct of the *Kdm2a*-dependent autosomal PRC2-related gene repression that we had identified at the preceding stage of differentiated spermatogonia. Consistently, the H3K36me2, H3K4me3 and H3K27me3 histone marks were marginally changed at the promoters of X-linked genes in SgD samples (Supplemental Fig.11a, 11c). Taken together, the inefficiency in the developmentally programmed downregulation of X-linked gene expression in *Kdm2a* iKO ScLZ along with the widespread retention of ATR and γ H2A.X on chromatin in *Kdm2a* iKO pachytene-like spermatocytes identifies KDM2A as a potentially direct or indirect regulator of DNA damage response-induced gene silencing (30).”

4. I would also like to see more of the discussion of the role of H3K36me2/me1 in transcription activation/repression/DNA integrity.

Thank you for your comment. We have revised the text to accommodate more discussion around the interplay of H3K36me2 and DNA methylation in page 18 line 467-480:

“Loss of KDM2B expression in ESCs resulted in aberrant DNA methylation at certain CGIs controlled by PRC1 (39). In the male germline, *de novo* DNA methylation is mainly facilitated by DNMT3A, which potentially interacts with H3K36me2 through its PWWP domain. When NSD1 is depleted, H3K36me2 levels decrease, leading to a reduction in DNA methylation and an expansion of Polycomb-mediated gene silencing (23). In *Kdm2a*-mutant spermatogonia, CGI-rich promoters may resist abnormal gains of DNA methylation due to their generally high occupancy of H3K4me3, which is known to counteract the binding of the *de novo* DNA methyltransferases (10,40). Consistently, gene promoters in cl. 5 exhibiting the lowest H3K4me3 marking among CGI-rich gene promoters showed the highest gain of DNA methylation. In addition, the presence of KDM2B in *Kdm2a*-deficient spermatogonia could sustain a basal level of H3K36 demethylation activity, which would prevent excessive gain of DNA methylation. Further investigation of simultaneous depletion of *Kdm2a* and *Kdm2b* is required to shed light on the interplay of H3K36me2 and DNA methylation at gene promoters during spermatogenesis.”

Reviewer #3 (Remarks to the Author):

In this manuscript, the authors aimed to demonstrate the essential role for the histone H3 lysine demethylase KDM2A/FBXL11 in spermatogenesis during development in both the tamoxifen inducible and the constitutive Ngn3-Cre driven knockout mouse models. The authors used multiple experimental approaches, including histological and molecular methods, characterized the spermatogonial phenotypes in great details, and well-documented the spermatogonial defects in those conditional knockout mice. They present a series of interesting new findings and propose a potentially important model of this biologically important event. The manuscript is well written and the data included are very clear, there are several serious concerns that need to be addressed before

publication. Specific points are listed below.

1. KDM2A has been known to play an essential role in mouse embryonic development by regulating cell proliferation and survival (Kawakami et al, 2015). Therefore, it is not so surprising to know that *Kdm2a* deficiency in mice causes defects in spermatogonial cell proliferation and differentiation, delayed cell cycle progression and apoptosis. What remains largely unknown is the underlying molecular mechanisms. It appears that KDM2A plays an important role in controlling expression of numerous genes that are responsible for spermatogonial differentiation and progression through meiotic prophase. However, the study only scratched the surface with little mechanistic insight.

We appreciate the reviewer's suggestion to provide further mechanistic insight regarding the differential gene expression. Considering that many adult tissues rely on cell proliferation and differentiation for their normal function, it is actually surprising that the *Kdm2a* mutation has such a pronounced effect specifically in spermatogonial cells only. Moreover, given that the JmjC domain mutation in the *Kdm2a* paralog, *Kdm2b* (Ozawa, et al., 2016), does not result in spermatogenic arrest, the dramatic phenotype associated with *Kdm2a* is particularly surprising.

To gain a deeper understanding of how KDM2A influences gene expression, we performed ChIP sequencing to assess the levels of its substrate, H3K36me2, at gene promoters in differentiated spermatogonia. Our analysis revealed that, in the absence of *Kdm2a*, all CGI-rich and some CGI-poor promoters exhibited an increase in H3K36me2. Additionally, we examined activation-related chromatin modification H3K4me3, repression-related H3K27me3, and DNA methylation marks to determine what sensitizes genes to respond to the *Kdm2a* depletion-associated gain in H3K36me2. We found that the CGI-rich upregulated genes in *Kdm2a* mutant differentiated spermatogonia have been already expressed in low to mid-levels in undifferentiated spermatogonia and failed to be repressed upon differentiation because they are:

1) marked by low levels of H3K27me3 and intermediate levels of H3K4me3 in the control condition and tend to gain H3K4me3 in *Kdm2a* mutant which overcomes the PRC2 repression. See figure 5 and Supplementary figure 9 for genes in cluster 2.

2) marked by intermediate levels of H3K27me3 and intermediate levels of H3K4me3 in the control condition and lose the H3K27me3 in *Kdm2a*, accompanied by a gain in H3K4me3 which eventually again overcomes the PRC2 repression. See figure 5 and Supplementary figure 9 for genes in cluster 3 and some genes in cluster 4.

For CGI-poor genes in cluster 7 which show upregulation in *Kdm2a* mutant differentiated spermatogonia, they were neither expressed in undifferentiated spermatogonia, nor exhibit changes in the H3K4me3 or H3K27me3 modifications between control and *Kdm2a* mutant differentiated spermatogonia. These genes exhibit only a marginal increase and most of them were excluded from the statistical analysis due to their low expression. Currently we do not have a mechanistic explanation as to how these genes are slightly upregulated. We have entitled the result section "KDM2A is required for silencing of PRC2 target genes in differentiating spermatogonia" and Figure 5 and Supplemental Figure 9 to present these novel findings.

2. KDM2A is a histone demethylase specific for mono- and di-methylated histone H3 lysine 36. However, in this manuscript the authors did not examine changes in this histone mark in spermatogonial cells at different developmental stages upon loss of *Kdm2a*.

Thank you for pointing out this omission. In the revised version of this manuscript, we have determined the relative level and genomic distribution of the preferred substrate of KDM2A, H3K36me2 by performing a ChIP-seq experiment optimized for small input samples (ULI-NChIP-seq assays). We present this data in Figure 5a and 5c as well as in Supplemental Figure 9. Our findings regarding the changes of H3K36me2 at gene promoters are presented in the Results section page 13 line 328-336:

“Genes highly expressed throughout spermatogenesis, which were found in cl. 1 and 8, associated with housekeeping and reproduction functions respectively (Supplemental Fig. S9a) and showed minimal transcriptional changes between Ctrl and *Kdm2a* iKO cells (Fig 5b). CGI-positive gene promoters in cl. 1 showed depletion of H3K36me2 and high enrichment for H3K4me3 in both Ctrl and iKO SgD (Fig. 5a, 5c and Supplemental Fig. 9c). Conversely, CGI-negative gene promoters in cl. 8 were devoid of H3K4me3 and exhibited higher levels of H3K36me2 compared to cl. 1 gene promoters in both Ctrl and *Kdm2a* iKO SgD cells, arguing that the gene and chromatin regulation of highly expressed genes was largely *Kdm2a* insensitive (Fig. 5a, 5c and Supplemental Fig. 9c).”

And page 13 line 343-346:

“However, only cl. 7 genes gained H3K36me2 at their promoters in *Kdm2a* iKO compared to Ctrl SgD, which may lead to their marginal upregulation, despite persistent levels of DNA methylation (Fig. 5a, 5c, and Supplemental Fig. 9c).”

And page 13 line 351-353:

“In *Kdm2a* iKO SgD, gene promoters in cl. 2, 3 and 4 showed increased H3K36me2, in particular around their TSSs (Fig. 5a, 5c, and Supplemental Fig. 9c, 9d, 9f), arguing for being direct targets of KDM2A.”

3. The authors performed RNA-sequencing assays using RNA from purified spermatogonia and spermatocytes, and identified groups of candidates. Unfortunately, the data included in the paper are largely descriptive with no validation of selected candidates. For example, while *kdm2a* is primarily required for gene repression, the authors listed one group of genes in Figure 6 that failed to be induced in iKO cells, but did not perform any further validation of expression of these genes, and simply mention “their delayed or impaired activation might underlie the impairment of meiotic progression and cell death observed for *Kdm2a* iKO spermatocytes”. It would be interesting and very important to know whether differential gene expression is correlated with local changes in the level of methylated H3K36 in *Kdm2a* deficient cells.

Thank you for your comment. We have repeated our RNA sequencing experiment and have produced in-house ChIP-seq data for H3K36me2 and other histone modifications (see above) in Ctrl and *Kdm2a* iKO differentiated spermatogonia (SgD). This is the male germ cell population that we have detected the most widespread gene expression differences between Ctrl and *Kdm2a* iKO animals. Based on our newly generated in-house chromatin datasets, we have regenerated the clustering of the genes. Differentially expressed genes are preferentially found in clusters 2, 3, 4 and 7 (Fig. 5a and 5b). Their promoters show increased signal for H3K36me2 (Fig. 5a, 5c, Supplemental Figure 9c, 9d) in *Kdm2a* iKO SgD.

4. It has been shown that KDM2A is ubiquitously expressed during embryogenesis and in several adult tissues (Fukuda et al, 2011; Kawakami et al, 2015). Whether or not the same is true for KDM2A in spermatogonial tissue remains unclear. It would be very helpful if the authors could show the expression pattern of endogenous KDM2A during proliferation and differentiation of spermatogonia.

Thank you for your comment. We have performed immunofluorescence staining of testicular section with anti-KDM2A specific antibody to study the expression pattern of KDM2A during spermatogenesis. We have included representative images in Figure 1f and Supplemental Figure 2 and summarized our findings in Figure 1f. We have revised the Result's section page 6 line 132-148:

“Immunofluorescence (IF) analysis of testicular sections showed that the KDM2A protein labels the entire nuclei of undifferentiated spermatogonia strongly, and the nuclei of differentiating spermatogonia, and of leptotene and zygotene spermatocytes more moderately (Fig. 1f, and Supplemental Fig. S2). In nuclei of pachytene and diplotene spermatocytes and of early round spermatids, KDM2A becomes enriched at DAPI bright regions containing pericentromeric major satellite sequences packaged in a *Suv39h1/2*-dependent constitutive heterochromatin state (Fig. 1f, 1g and Supplemental Fig. S2) (13). After 9 days following tamoxifen treatment, all these germ cell populations exhibited greatly reduced levels of KDM2A in *Kdm2a* iKO testes (Fig. 1f and Supplemental Fig. S2).”

5. The authors provide clear in vivo evidence showing that KDM2A is indispensable for the generation and maintenance of differentiating spermatogonia. Given that the culture of spermatogonial stem cells of mice was proved to be successful (Kanatsu-Shinohara M, 2007), it may be very useful if the authors could confirm the in vivo data by ex vivo culture of the spermatogonial stem cells from control and *Kdm2a* deficient mice.

Thank you for your suggestion. In our manuscript, we claim that KDM2A is indispensable for the maintenance of spermatogonia stem cells, which are part of undifferentiated spermatogonia. The specification/generation of spermatogonia stem cells from prospermatogonia/gonocytes has already commenced during the perinatal period of germ cell development, before the induction of Cre driven recombination in either of our models (tamoxifen- or *Ngn3*- induced). We have carefully considered your proposal to test a *Kdm2a* deletion model in spermatogonial stem cells cultures, we believe that the effort and resources required would not yield any significant advancement to our understanding about the role of *Kdm2a* during spermatogenesis. Below we list reasons why we recommend against conducting this experiment:

1) For this project, during the pre- and post- submission phases, we developed novel tools and experimental procedures to investigate the *Kdm2a* function in vivo. Specifically, we developed a) a novel FACS sorting protocol to isolate three differentiation phases of male germ cells and b) sensitive RNAseq and ULI-NChIP-seq assays that allowed us to use limited amount of starting material to perform transcriptomic and epigenomic analysis.

2) In vivo, KDM2A is required for gene regulation (in particular transcriptional repression) during the spermatogonia differentiation. The spermatogonia stem cells culture model will be suboptimal to capture this function, since there are no defined conditions that enable spermatogonia stem cells to be synchronously differentiated in vitro.

6. In page 4 line 106, the authors need to mention which Cre driver was used to generate the homozygous *kdm2a1flox/1flox* mice that died at mid-gestation.

Generation of the *Kdm2a* 1lox/1lox mice that died at mid-gestation, was achieved by crossing *Kdm2a* 2lox/2lox mice with transgenic mice that express Thy1-Cre recombinase in the germ line in an approach used by Lohman et al (PMID: 20014010). This particular Cre recombinase line did not exhibit the anticipated tissue-specific expression; instead, its activity was detected at very early developmental stages. Consequently, it proved effective as a germline deletion tool.

7. In Fig 1d, it would be better to replace the absolute weight with normalized weight (normalized by body weight).

Thank you for your suggestion. With respect, we do not share the reviewer's opinion and consider it more informative to present absolute testis weights (unless mutant mice would be runt). In the Supplemental Figure 1a we provide evidence that the total body weight of the *Kdm2a* iKO animals remain unchanged, while the *Ctrl* animals slightly gain weight during the experimental course. Despite the significance of these measurements, the fold changes in weight gain of *Ctrl* versus *iKO* animals are marginal (1.1-fold at 9 weeks post tamoxifen treatment). Normalization to the total body weight will not change the conclusion stated on page 5 line 132-134:

"The only apparent phenotype we observed in *Kdm2a* iKO mice involved the testes, which were markedly smaller and displayed an approximately 80% decrease in absolute weight (Fig. 1c, 1d)."

Reviewer #2 (Remarks to the Author):

The revised manuscript by Bocker *et al.*, describes the phenotypic and molecular characterization of conditional mouse knockout of the histone H3 lysine 36 demethylase KDM2A. The authors convincingly demonstrate that the knockout has a severe defect during spermatogenesis and suggest that this is a consequence of inappropriate expression of a class of PRC regulated genes. The authors have done an excellent job addressing the previous reviews, including new experimental data and in the presentation. This is an important study with implications for understanding how spermatogenesis is regulated at the transcriptional level and will be of interest to the broad audience of Nature Communications.

We sincerely appreciate your positive feedback and your recognition of the significance of our study for the readership of Nature Communications.

Reviewer #4 (Remarks to the Author):

The manuscript entitled “The histone H3 lysine 36 demethylase KDM2A/FBXL11 controls PRC2-mediated gene repression and germ cell development in male mice” by Michael *et al.* exploited both germline-specific and tamoxifen-induced mouse models to explore the roles of Kdm2a, a demethylase for H3K36me2/1, in the germline development in mouse testicular development. They found that Kdm2a serves an important role during the differentiation of spermatogonia. Mechanistically, Kdm2a repress the PRC2-target gene expression to safeguard the correct spermatogonia-to-meiotic transition. This study is overall well-designed, the results are clearly described with sufficient mechanistic evidence. The data strongly support the major conclusions. I believe it is a significant study in the male reproductive field. I have only minor comments as below:

We appreciate your kind words regarding the study's design, clarity, and mechanistic evidence. We are grateful for your recognition of the importance of our findings in the context of male reproductive biology. Your comments are highly valued, and we addressed them carefully in the revised manuscript to further improve the quality of the work.

1. With the easy access to the available single-cell RNA-seq datasets, please also include a plot showing the dynamic expression levels of KDM2a mRNA throughout germline development. It would be interesting to see whether Kdm2a is much higher in differentiating spermatogonia since it is functional at this stage.

Thank you for the insightful suggestion. We have analyzed single-cell RNA-seq from the adult murine spermatogenesis published in: “The mammalian spermatogenesis single-cell transcriptome, from spermatogonial stem cells to spermatids” by Hermann BP, *et al.* <https://doi.org/10.1016/j.celrep.2018.10.026> (1). Using the clustering and cell type annotations provided by Hermann BP, *et al.*, we plotted the expression of *Kdm2a* and its paralogue *Kdm2b* across various spermatogenic and somatic cell types, which we present in Supplemental Figure 2a. In addition, we added in the result section page 6 line 148:

“To examine the expression pattern of *Kdm2a* and its paralogue *Kdm2b* during spermatogenesis we reanalyzed previously published single-cell RNA sequencing data from

adult mouse testes (13) (Supplemental Fig. 2a). *Kdm2a* and *Kdm2b* are highly expressed in a large fraction of premeiotic spermatogonia and expressed at lower levels from pachytene spermatocytes onwards. Additionally, *Kdm2a* expression was detected in somatic Sertoli cells.”

2. Could the authors explain why the c-kit signal was so strong in the interstitial cells in the testes in the immunofluorescence staining images, e.g. Figure 1F, Supple Fig.2 ?

Thank you for the question. In the past, multiple groups have demonstrated high expression of c-KIT in the Leydig cells:

- Gonadal expression of *c-kit* encoded at the *W* locus of the mouse. Manova, *et al.* Development (1990) <https://doi.org/10.1242/dev.110.4.1057> (2)

In Figure 8E, the authors show RNA *in situ* hybridization with *c-Kit* antisense probe which shows expression outside of seminiferous tubules in Leydig cells.

- Differential Expression of *c-kit* in Mouse Undifferentiated and Differentiating Type A Spermatogonia. Schrans-Stassen, *et al.* Endocrinology (1999) <https://doi.org/10.1210/endo.140.12.7172> (3)

In Figure 2 the authors show immunolocalization of c-KIT protein at Leydig cells.

- Stem cell factor functions as a survival factor for mature Leydig cells and a growth factor for precursor Leydig cells after ethylene dimethane sulfonate treatment: implication of a role of the stem cell factor/c-Kit system in Leydig cell development. Yan, *et al.* Developmental Biology (2000) <https://doi.org/10.1006/dbio.2000.9885> (4)

In Figure 8 the authors show immunohistochemical detection of c-KIT on precursor Leydig cells and they additionally argue that the SCF -the ligand of c-KIT- functions as a survival factor for mature Leydig cells.

In addition, our reanalysis of single cell RNA seq data from Hermann, *et al.* (1) showed that around 50% of Leydig cells show the highest expression of *c-Kit*, followed by high levels of *c-Kit* in around 60% of differentiating spermatogonia (Rebuttal Figure 1).

Rebuttal Figure 1. Boxplots showing the log₂-normalized expression of *c-Kit* in different spermatogenic and somatic cell populations as defined by Hermann *et al.* (1). The number and the percentages of cells belonging assigned to a cell type expressing the *c-Kit* are shown above the boxplots.

In agreement with the public scRNA expression data, the intensity of c-KIT staining is lower in spermatogonia compared to the Leydig cells in our immunofluorescence figures (Figure 1F and Supplemental Figure 2B). We have updated Figure 1F by adding dashed lines to delineate the contour of the seminiferous tubules and asterisks pointing to Leydig cells. We envision that this labeling will assist the readers to differentiate between intra-tubular c-KIT positive spermatogonia and interstitial c-KIT positive Leydig cells.

Importantly, during the preparation of the germ cell suspensions for RNA-seq, Leydig and other interstitial cells were effectively depleted by isolating and thoroughly washing the seminiferous tubules before cell dissociation. This is reflected in the low expression of Leydig cell specific gene markers in our RNA sequencing data, as shown in Supplemental Figure 6G.

3. For calculation of exon expression levels in supple Fig.6B. the authors mentioned that there was a 35% reduction in the total read numbers in the iKO germ cells. Were

these calculated against averaged exon levels ? we usually observed the compensatory increase in the levels of the adjacent exons around the floxed exons in the conditional KO models from the RNA-seq data. This could be checked by showing a snapshot of the IGV track of the RNA-seq data.

Thank you for your comment. We would like to clarify here how the exon expression levels in Supplemental Figure 6B were calculated.

We calculated for each cell type the median RPKM values of 4 biological replicates from either iKO or Ctrl samples for *Kdm2a* exons 1-7 and 9- 21 (excluding the floxed exon 8). We then calculated the percentage of the reads in iKO divided by the reads in Ctrl as shown in the table below.

Kdm2a exons 1-7 and 9-21 (data for Supplemental Figure 6B)

Cell type	Genotype	Median RPKM	% reads (iKO/Ctrl)*100
SgU	Ctrl	9	
SgD	Ctrl	13.5	
ScLZ	Ctrl	11.7	
SgU	iKO	6.17	68.55
SgD	iKO	8.37	62
ScLZ	iKO	8.03	68.63

Thus, the reduction of exons 1-7 and 9-21 altogether spans from 32% to 38%.

We then calculated for each cell type the median RPKM values of 4 biological replicates from either iKO or Ctrl for *Kdm2a* exon 8 – i.e. the floxed exon. We then calculated the percentage of the reads in iKO divided by the reads in Ctrl as shown in the table below.

Kdm2a exon 8 (data for Supplemental Figure 6C)

Cell type	Genotype	Median RPKM	% reads (iKO/Ctrl)*100
SgU	Ctrl	7.37	
SgD	Ctrl	8.53	
ScLZ	Ctrl	12.6	
SgU	iKO	0.928	12.59
SgD	iKO	1.19	13.95
ScLZ	iKO	2.66	21.11

Thus, the reduction of exon 8 spans from 79% to 88%.

We additionally provided the requested UCSC browser snapshot for the *Kdm2a* locus with RNA-seq tracks in Supplemental Figure 6D. In this snapshot, the neighbor exons 7 and 9 do not show major expression changes. Overall, we did not notice any compensatory upregulation of the non-floxed exons as shown in Supplemental Figure 6B. Instead, the marked decrease of the non-floxed exons suggests that the transcripts generated without exon 8 are overall less stable compared to the wild type.

4. In Supple Fig.9F, have you ever compared the H3K36me2 peak intensity between Ctrl and iKO in 10 day SgD ? It appears that the peak intensity levels are comparable

across the gene bodies when comparing Ctrl with iKO. A zoom-in view of IGV track in the promoter regions would be more informative to the readers. In addition, have the authors exploited the CUT&Tag to do the H3K36me2 modification? It has been reported to be powerful to identify the promoter region alongside the gene body localization (<https://doi.org/10.1038/s41467-022-30940-1>)

Thank you for raising this interesting point. We have previously compared the H3K36me2 signal at and around gene TSS +/-5kb as shown now in Main Figure 5D (previously Supplemental Figure 9C). We observed depletion (aka a valley) of H3K36me2 instead of a peak at the TSS in the Ctrl samples, presumably due to the function of KDM2A removing this histone mark from the TSS. This observation is in line with the signal of H3K36me2 at TSS in mouse embryonic stem cells reported by Turberfield *et al.*, 2019 (<https://doi.org/10.1093/nar/gkz607>) (5). In the iKO samples the valley gets shallower.

To address the reviewer's point, we next interrogated a wider set of genomic regions, -5kb to +25kb relative to TSS, effectively incorporating gene bodies of genes shown in the clusters of Figure 5 (Rebuttal Figure 2).

Rebuttal Figure 2. Line plots showing average H3K36me2 \log_2 RPKM values along genes (-5kb to +25kb relative to transcriptional start sites (TSS)) belonging to clusters 1-9 as defined in Figure 5, and assessed by ULI-NChIP-seq from Ctrl (*Kdm2a*^{2lox/2lox}) and *Kdm2a* inducible knockout (iKO: *Kdm2a*^{2lox/2lox}; CAGG-Cre-ERTM) (n=2 for H3K36me2 from each genotype) animals 10d post first tamoxifen treatment.

Despite the relative increase of H3K36me2 at almost all TSS in iKO samples compared to Ctrl samples, we observed very similar levels of H3K36me2 at gene bodies in both genotypes which is in agreement with the reviewer's observation regarding the comparable levels of H3K36me2 at the 3 example loci of Supplemental Figure 9F. In order to highlight the effect of iKO primarily at the gene TSS we provided zoom-up views of promoters of the genes as part of Supplemental Figure 9F.

We have not performed CUT&Tag ourselves, but we reanalyzed publicly available a-H3K36me2 CUT&Run (C&R) data from Feng *et al.*, 2024 (<https://doi.org/10.1038/s44318-024-00203-4>) (6) and compared these data to the a-H3K36me2 ULI-NChIP data from our study.

As a first quality control, we investigated potential GC biases and calculated \log_2 RPKM values for 1kb genomic tiles of the C&R or ULI-NChIP data. We then plotted these values versus GC% of the interrogated genomic tiles. We noted that particularly the Ctrl samples (both input and H3K36me2 C&R) from the Feng *et al.* study showed higher GC bias compared to their KO samples and to our ULI-NChIP samples (Rebuttal Figure 3).

Rebuttal Figure 3. A) Scatterplots showing the \log_2 RPKM values of 1kb genomic windows (100k randomly chosen windows) versus GC percentage of the windows from C&R replicate samples from Feng et al for a-H3K36me2 and input in Ctrl and KO samples. Windows overlapping with CGI are marked red. Colorcode denoted the density of the point on the scatterplot. B) As in A showing the samples for a-H3K36me2 ULI-NChIP from the current Bocker et al study.

Since many TSS sites are characterized by high GC percentages (see genes in clusters 1-5 of Figure 5), such GC bias effect needs to be normalized to mitigate artifacts. Therefore, after having merged the replicates of each experimental condition, we performed loess normalization of all C&R samples (a-H3K36me2 and inputs of control and mutant genotypes) and then normalized the H3K36me2 data to the input. This approach greatly reduced the GC biases compared to the non-normalized data (Rebuttal Figures 3 and 4).

Rebuttal Figure 4. A) Top 4 scatterplots showing loess normalized \log_2 RPKM values of 1kb genomic windows (presented are 100k randomly chosen windows) versus GC percentage of the windows from C&R merged replicates from Feng *et al.* for a-H3K36me2 and input in Ctrl and KO samples (top plots). Bottom 2 scatterplots showing ratio of loess normalized \log_2 RPKM values of merged C&R a-H3K36me2 values over input values for 1kb genomic windows (100k randomly chosen windows) versus GC percentage of the respective genomic windows.

B) Scatterplots showing loess normalized \log_2 RPKM values of 1kb genomic windows (presented are 100k randomly chosen windows) versus GC percentage of the windows from merged replicates from for a-H3K36me2 ULI-NChIP Ctrl and iKO samples from the current study.

We used this approach to normalize the levels of a-H3K36me2 assayed by C&R and by ULI-NChIP at TSSs following our clustering of Figure 5. Data of H3K36me2 C&R that is not normalized for input values show enrichment at all TSS (Rebuttal Figure 5). Such enrichment of not input normalized H3K36me2 is presented by Feng *et al.* in their Figure 5J plot.

However, given that their input samples also showed marked enrichment at TSS, this potentially reflects the GC biases of these libraries. Hence, when we normalized their H3K36me2 signal to their input, we observed clear depletion of H3K36me2 at TSS, comparable to our ULI-NChIP seq data presented in Main Figure 5A and 5D (Rebuttal Figure 5). In conclusion, the C&R and ULI-NChIP data capture similar chromatin signatures of H3K36me2 at TSS when proper normalization has been applied.

A

Rebuttal Figure 5. A) Heatmap showing chromatin signatures of 5kb regions upstream/downstream of their corresponding TSSs. All genes were classified into 9 clusters by hierarchical clustering using “Ward.D” agglomeration method on a dissimilarity matrix of “Euclidean” distance based on gene RNA expression values and chromatin modifications measured by ULI-NChIP-seq. and shown in Figure 5A.

From left to right: Cluster identity, Chromosomal location of the gene, Presence or absence of CpG island from the gene promoter, CpG density at gene TSS +/- 5kb, log₂ RPKM values of H3K36me2 ULI-NChIP-seq for non-overlapping 50bp genomic window located +/- 5kb from TSSs in Ctrl and iKO SgD samples of the current study, log₂RPKM values of input, H3K36me2 and H3K36me2/input C&R for non-overlapping 50bp genomic window located +/- 5kb from TSSs in Ctrl and KO samples from Feng *et al.*.

B) Line plots showing average input and H3K36me2 (top) and H3K36me2/Input (bottom) log₂RPKM values at TSSs +/- 5kb belonging to the clusters 1-9 as defined in Figure 5 assessed by C&R from Feng *et al.*

C) For direct comparison: Current Figure 5D.

Reviewer #5 (Remarks to the Author):

In this study, the authors describe a critical role for the histone demethylase KDM2A in supporting adult spermatogonial differentiation and meiotic progression.

Mechanistically, they performed multiple omics analyses on isolated spermatogonia and spermatocytes and demonstrated that KDM2A is required for PRC2-mediated repression by increasing H3K36me2 and decreasing H3K27me3 promoter occupancy. Furthermore, they showed that Kdm2a ablation in spermatocytes disrupts meiotic progression, accompanied by impaired chromosome synapsis completion and DSB processing, likely due to altered chromatin states and impaired X-linked gene repression. However, the main criticism of the study is its novelty, as a similar study has already revealed the function of KDM2A in meiosis (PMID: 39160277).

We would like to acknowledge the effort and time of the reviewer to provide a constructive criticism of our work. During the long process of the first round of revision, a paper arguing that the testis-specific knock out of *Kdm2a* is nonessential for male

fertility was published by Xiong and colleagues (doi.org/10.1016/j.theriogenology.2023.06.008) (7). More recently, Feng and colleagues (doi.org/10.1038/s44318-024-00203-4) (6) published a paper arguing that KDM2A orchestrates the entry and progression of male germ cells into meiosis by regulating early meiotic gene expression as measured in juvenile mice.

Going beyond these two reports, we identified novel and critical roles for KDM2A in adult mice in coordinating Polycomb-dependent changes in specific gene expression programs that normally occur during spermatogonial differentiation which are essential for male germ cell development. Contrasting Feng *et al.*, we characterized KDM2A as a H3K36me2 demethylase critical for gene repression during spermatogonial differentiation. Moreover, developmentally speaking, we identified differentiating spermatogonia as the earliest population of male germ cells affected by *Kdm2a* deletion. Our study demonstrates the important regulatory function of KDM2A in shutting down expression in differentiating spermatogonia prior to entry into meiosis, which is key for germ cell development. Finally, our findings point to an important role for KDM2A in meiotic sex chromosome inactivation, a process that is crucial for completion of meiosis and hence male fertility.

Here, we provide a comprehensive comparative analysis of our and Feng studies. Both studies report reduced testis weight, infertility, and defects in meiosis upon depletion of KDM2A in mouse spermatogenesis. However, due to experimental differences among the studies' design concerning the different timing between the Cre induced recombination and the analysis of the samples, the two reports investigated different aspects of the phenotype. Accordingly, Feng *et al.* focused on the juvenile period of spermatogenesis using the *Stra8*-CreGFP driven *Kdm2a* deletion acting during the first wave of spermatogenesis. Feng *et al.* observed that P10 c-KIT⁺ cells showed downregulation of genes involved in meiotic functions and thereby hypothesized that meiotic initiation is compromised. To support this, they reported decreased RAD51 and DMC1 foci marking the double strand breaks. In our study, we performed a tamoxifen inducible time course evaluation of adult spermatogenesis and found that after 12d post injection the spermatocytes which have normally entered meiosis display meiotic synapsis defects and persistent phosphorylation of γH2AX on autosomes.

Moreover, both studies identified a decrease in the number of spermatogonia either at 35 d post injection in the *Ddx4*-CreERT2 or after 9 d in the *Actb*-CreERT2 models. In our model we identified a plethora of upregulated genes which are associated to developmental functions similar to the upregulated genes that Feng *et al.* reported in their P10 c-KIT⁺ cells. Both studies showed that upregulated genes are highly expressed in spermatogonia stem cells. We showed that these genes are normally downregulated during the course of spermatogonial differentiation and reported that H3K36me2 is depleted, while H3K27me3 is enriched at the TSS of such genes in control cells. In our mutant model, we reported an increase of H3K36me2 occupancy, which presumably reflects a catalytic role of KDM2A at such genes. We further associated the upregulation of H3K36me2 with a defective H3K27me3 deposition, thus implicating KDM2A in defining the PRC2 target repertoire.

Feng *et al.* showed that KDM2A binds to the TSS of both upregulated and downregulated genes and reported enrichment of both H3K36me2 and H3K36me3 at the gene TSS in the control cells, which is contradictory to the perceived role of the KDM2A in H3K36 demethylation. In our response to Reviewer #4 Comment 4 we show that the H3K36me2 and input Cut&Run data from Feng *et al.* suffer from high GC bias which after proper normalization show that H3K36me2 is actually depleted from TSS rather than being enriched.

Finally, an important point is that Feng *et al.*, 2024 reported that H3K36me2 is reduced at both up and down-regulated genes TSS in the mutant condition, while we showed that with proper data normalization of either a-H3K36me2 C&R performed by Feng *et al.* or a-H3K36me2 ULI-NChIP performed by our study, H3K36me2 shows marked increases in the majority of TSSs in *Kdm2a* ablated germ cells (Rebuttal Figure 5 and Main Figure 5A and 5D).

Other major points:

1. For Figure 1F, the cells labelled by the c-KIT antibody are probably not spermatogonia, but rather interstitial cells of the testis. Please specify the cell type of the KDM2A positive cells.

Thank you for your comment. Beyond differentiating spermatogonia, high levels of c-KIT are observed in interstitial Leydig cells. We kindly refer to our reply to Comment #2 of reviewer #4 for our extended explanation and data.

In order to facilitate the reading of Figure 1F, we updated this figure showing a seminiferous tubule stage VII with more c-KIT positive spermatogonia residing in the periphery of the seminiferous tubules. In addition, we delineated the border/periphery of seminiferous tubules and added asterisks to point where interstitial cells are localized. In Figure 1G we provide a schematic representation of KDM2A expression in various germ cells types based on our the immunofluorescence analysis, also shown in Supplemental Figure 2B. Regarding the expression of KDM2A in somatic cell types please see our response to your second comment below.

2. It is better to determine whether KDM2A is expressed in testicular cells (e.g. Sertoli cells and Leydig cells) in addition to germ cells. Although the authors found that the Actb-CreERT2-induced *Kdm2a* knockout mice had a similar phenotype to the germ cell-specific *Kdm2a* knockout mice driven by *Ngn3-cre*, it cannot be ruled out that KDM2A may play a role in other testicular cells for spermatogenesis.

To determine the gene expression of *Kdm2a* in testicular cells we reanalyzed publicly available single cell RNAseq data from Hermann *et al.* (1) and show in Supplemental Figure 2A that *Kdm2a* is also expressed in somatic testicular cells like Sertoli cells, endothelial cells and in Leydig cells.

Consistently, reviewing our KDM2A immunofluorescence staining, there is prominent KDM2A signal in Leydig cells (cKIT positive cells residing in the interstitial space) (Supplemental Figure 2B and Rebuttal Figure 6). Sertoli cell nuclei (residing close to the basal membrane with typical heterochromatic DAPI bright spot) also express low levels of KDM2A signal localized particularly at the DAPI bright heterochromatin. Sertoli cells at stage VII and I tubules, show higher euchromatin KDM2A signal compared to Sertoli cells at other stages (Supplemental Figure 2B). Finally, Myoid cell nuclei (forming the basal membrane with typical elongated nuclei) are devoid of KDM2A signal.

Rebuttal Figure 6. Immunofluorescence staining of Ctrl testicular sections with an anti-cKIT (green) and anti-KDM2A (red) antibodies. Nuclei were counter stained with DAPI (white). Myoid cell nuclei are encircled in red, Sertoli cell nuclei are encircled in green, Leydig cell nuclei are encircled in blue, Undifferentiated spermatogonia nuclei are encircled in purple, Differentiated spermatogonia nuclei are encircled in yellow and Spermatocyte nuclei are encircled in white. Scale bar = 10 μ m.

Based on these observations, we agree with the reviewer that KDM2A may have functional roles in Leydig and Sertoli cells. Yet, for Leydig cells, the iKO model is not suitable for studying KDM2A function, since protein levels remain largely unchanged and high in Leydig cells at day 9 post addition of tamoxifen (Figure 1F and Supplemental Figure 2B). In Sertoli cells, expression of KDM2A shows variable levels according to the seminiferous tubule stage in control samples (Supplemental Figure 2B). The effect of *Kdm2a* gene deletion on KDM2A expression in Sertoli cells is not fully penetrant with variability observed in the degree of reduction of KDM2A protein levels (which is related to stage of the seminiferous epithelium). Nevertheless, the ratio of Sertoli cells versus undifferentiated spermatogonia was not altered at day 65 post addition of tamoxifen (Figure 3E-F). Despite that, we cannot exclude that KDM2A deletion would impact the chromatin status (eg the H3K36 methylation levels) in the somatic testicular cells. Such analyses, however, go beyond the scope of this work. Importantly, the germ cell specific *Ngn3*-Cre model supports our conclusion that differentiating spermatogonia are intrinsically susceptible to loss of *Kdm2a* function.

3. Did abnormal meiosis occur in the *Kdm2a* sKO mice during the first wave of spermatogenesis, similar to that observed in *Kdm2a* iKO mice?

Thank you for your question. Regretfully we cannot provide at this moment a satisfactory response to your question. We used the *Ngn3*-Cre sKO model to answer whether the observed phenotype is germ cell intrinsic and for the rest of the analysis we have been using the *Actb*-CreERT2 or the CAGG-CreERTM iKO models. From the *Ngn3*-Cre sKO model we analyzed only 2 timepoints at P26 and P29. Unfortunately, these timepoints are much later than the first wave of meiosis, which typically starts at P8-P10, preventing us from reaching any informative conclusions regarding the first meiosis using the available samples. Currently the *Ngn3*-Cre sKO model is not available in our vivarium and rederivation would require a significant amount of time and resources. As such, this question remains to be answered in future studies of the role of *Kdm2a* in spermatogenesis.

4. It is still unclear whether KDM2A ablation affects the overall changes in histone modifications in germ cells, so it is necessary to perform IF analysis in testicular sections using H3K36me1/2, H3K4me3 and H3K27me3 antibodies.

Thank you for your comment. We have already presented immunofluorescence data in Supplemental Figure 13 (former Supplemental Figure 12) regarding the changes we observed in H3K4me3 and H3K9me2 levels in pachytene spermatocytes.

We have now expanded the immunofluorescence experiment to include H3K27me3, H3K36me3, and H3K36me2. We didn't observe global changes of these marks, neither in spermatogonia nor in spermatocytes of seminiferous tubule matched stages between Ctrl and iKO samples. Importantly the genomic data argue that H3K36me2 specifically increases at CGI- containing promoters and H3K27me3 only in a subset of Polycomb targets. We believe that immunofluorescence analysis lacks the resolution to capture these local changes. We present this data in a new Supplemental Figure 14 and we discuss the results accordingly in the revised manuscript. Unfortunately, we could not detect H3K36me1 signal using the commercial antibody from Millipore 07-548.

5. The authors identified 779 up-regulated genes associated with "morphogenesis" and "cell adhesion" in the KDM2A-ablated differentiated spermatogonia, explaining the abnormal spermatogonial differentiation phenotype. Are the "cell death" related genes such as *Tnf*, *Egfr* and *Myc* specifically upregulated in the differentiated spermatogonia of *Kdm2a* iKO mice? Are their protein expression levels altered?

Thank you for your comment. In order to assess the protein expression levels of upregulated targets in a cell context manner we performed immunofluorescence analysis of MYC, NOTCH1, POU5F1(OCT4) and EOMES proteins for which we had access to readily available antibodies and given the existence of literature describing functions for these proteins during spermatogenesis. More specifically, these targets have been already shown to be expressed in mouse spermatogonia by Kanatsu-Shinohara *et al.* (2016) (<https://doi.org/10.1101/gad.287045.116>) (8), Murta *et al.* (2013) (<https://doi.org/10.1371/journal.pone.0072767>) (9), Pesce *et al.* (1998) ([https://doi.org/10.1016/S0925-4773\(98\)00002-1](https://doi.org/10.1016/S0925-4773(98)00002-1)) (10) and Sharma *et al.* (2019) (<https://doi.org/10.7554/eLife.43352>) (11) respectively. Interestingly, Huang *et al.* (2013) (<https://doi.org/10.1371/journal.pone.0071213>) (12) showed that conditional activation of the NOTCH1 intracellular domain (NICD) in germ cells results in decreased testis weight, reduction of the spermatogonial differentiation marker NEUROG3 expression and increased apoptosis. In addition, Zheng *et al.* (2016) (<https://doi.org/10.1530/REP-16-0140>) (13) showed that *Stra8*-cre induced expression of *Pou5f1* in the male germ lineage disrupts spermatogonial differentiation in mice.

All antibodies were used at manufacturer's recommended dilutions according to the staining protocol in Material and Methods section.

We tried to detect MYC using 2 different antibodies: (a) Abcam ab32072 and (b) Abcam ab32. In control samples, while ab32 didn't show any signal, the ab32072 showed widespread signal in all germ cells contrary to previous reports and single cell RNA seq data (Rebuttal Figure 7A, B). The disagreement between the 2 immunofluorescence experiments and the existing literature precluded us from extracting meaningful conclusions.

To detect NOTCH1 we used 2 different antibodies: (a) Cell Signaling Technology # 3608, (b) intracellular NOTCH1 provided by the laboratory of Dr. Verdon Taylor (Nyfeler Y, Kirch RD, Mantei N, Leone DP, Radtke F, Suter U, Taylor V (2005) Jagged1 signals in the postnatal subventricular zone are required for neural stem cell self-renewal.

EMBO J 24:3504–3515 <https://doi.org/10.1038/sj.emboj.7600816>) (14). Here, the commercial antibody showed widespread signals contrary to previous observations while the homemade didn't show any signal. Hence, we cannot assess this target either (Rebuttal Figure 7C, 7D).

Finally, for detection of POU5F1 (OCT4) we used the Santa Cruz sc-8628 and for EOMES we used the Abcam ab23345. Anti-POU5F1 (OCT4) staining showed very high background levels in particular in the cytoplasmic and extracellular spaces. Weak signals could be detected in the nuclei of undifferentiated spermatogonia (marked by green asterisks) in both Ctrl and iKO samples (Rebuttal Figure 7E). EOMES showed prominent nuclear staining in many germ cell types (spermatogonia and spermatocytes) in contrast to its previously reported exclusive expression in type As and Apr spermatogonia (11). The dynamics of presumable EOMES staining patterns appeared similar to between Ctrl and iKO (Rebuttal Figure 7E).

In summary, we cannot draw conclusions to support or reject the hypothesis that protein levels of upregulated gene targets are altered in differentiating spermatogonia. The challenges of antibody availability to all upregulated targets and the inconsistencies between antibodies staining and existing literature cannot be further addressed as part of this study.

A MYC-tag (Abcam ab32)

B c-Myc (Abcam ab32072)

C Notch1 (Cell signaling #3608)

D Notch1 (homemade from Verdon Taylor group, Nyfer et al., 2005)

Rebuttal Figure 7. A) Immunofluorescence staining of Ctrl testicular sections with an anti-cKIT (red) and anti-Myc-tag Abcam ab32 (green) antibodies.

B) Immunofluorescence staining of Ctrl testicular sections with an anti-cKIT (red) and anti-Myc Abcam ab32072 (green) antibodies.

C) Immunofluorescence staining of Ctrl testicular sections with an anti-cKIT (red) and anti-NOTCH1 Cell Signaling Technology # 3608 (green) antibodies.

D) Immunofluorescence staining of Ctrl testicular sections with an anti-cKIT (red) and anti-NOTCH1 Verdon Taylor group, Nyfeler Y et al., 2005 (green) antibodies. The acrosome and nucleus were counter stained with Lectin PNA (blue) and DAPI (white) respectively. Scale bar = 50µm.

E) Immunofluorescence staining of Ctrl (upper) and iKO (bottom) testicular sections with an anti-OCT4 and anti EOMES antibodies, anti-Myc-tag Abcam ab32 (green) antibodies. The acrosome and nucleus were counter stained with Lectin PNA (blue) and DAPI (white) respectively. Green asterisks mark DAPI-pale undifferentiated spermatogonia (As, Apr). Red arrows mark differentiating spermatogonia (A1-A4), with DAPI bright foci and DAPI bright nuclear periphery. Yellow dashed lines encircle late differentiating spermatogonia (In and B). Scale bar = 10µm.

6. For Figs. 5A, 5B, how many genes in cl.2 to 4 and cl.7 are included in the upregulated gene sets shown in Fig. 4E? Since the authors conclude that KDM2A is required for PRC2-mediated repression, why did only cl.3 genes show reduced H3K27me3 abundance?

Thank you for asking these important questions. We identified the most upregulated genes in the SgD iKO vs Ctrl contrast. As shown in the rebuttal figure 8, 194/778 are found in cluster 2, 251/778 in cluster 3, 95/778 in cluster 4 and 127/778 in cluster 7.

Rebuttal Figure 8. Stacked barplots showing percentages and absolute numbers of differentially expressed genes in each cluster for the 3 comparisons (SgU iKO vs Ctrl, SgD iKO vs Ctrl, ScLZ iKO vs Ctrl).

Regarding the second question, the Namekawa group showed that both PRC1 (Hu *et al.* 2024, <https://doi.org/10.1093/nar/gkad1203>) (15) and PRC2 (Maezawa *et al.* 2018, <https://doi.org/10.1073/pnas.1804512115>) (16) related histone modifications exhibit dynamic changes during spermatogenesis. We have re-analyzed both their H3K27me3 (from Maezawa *et al.*, 2018) and H2AK119u1 (from Hu *et al.*, 2024) occupancy levels at promoters following the clustering of genes displayed in Figure 5. We observed that gene promoters in cluster 4 are stably and highly enriched for both marks in every cell type examined by the Namekawa group (Rebuttal Figure 9). Cluster 2 and 3 gene promoters showed a gain in H2AK119u1 levels in differentiated type A spermatogonia relative to undifferentiated type A spermatogonia (Rebuttal Figure 9 and Supplemental Figure 10 B 10D). In addition, while cl. 2 and 3 gene promoters showed similar levels of H3K27me3 in THY1+ and cKIT+ spermatogonia of juvenile mice, they showed an increase of H3K27me3 signal in pachytene spermatocytes from adult mice.

We hypothesized that the specific transcriptional effects which we observe at cluster 3 genes upon KDM2A depletion in SgD cell may relate to the developmental changes in levels of H3K27me3 occurring during the subsequent stages of spermatogenesis (i.e from undifferentiated spermatogonia SgU, to SgD and Spermatocytes).

As the cell types that were isolated by Namekawa group to study the dynamics in H3K27me3 were obtained from juvenile animals (Maezawa *et al.*, 2018) and do not recapitulate exactly our adult FACS sorted germ cell populations, we decided to perform a comprehensive H3K27me3 profiling of adult wild-type FACS sorted germ cells in 4 different spermatogenic stages: undifferentiated spermatogonia (SgU), differentiated spermatogonia (SgD), Leptotene/Zygotene spermatocytes (ScLZ) and Pachytene /Diplotene spermatocytes (ScPD). As observed before, cluster 4 gene promoters exhibited persistently high levels of H3K27me3 during spermatogenesis. In contrast, H3K27me3 levels remained low for cluster 2 gene promoters in SgU, SgD and ScLZ, and only gained moderate levels of H3K27me3 during meiosis in ScPD. Finally, cluster 3 gene promoters showed progressive gain of H3K27me3 already at the transition from SgU to SgD and then further in ScLZ and ScPD spermatocytes (Supplemental Figure 10A, 10D and Main Figure 5A).

Hence, our novel data extend the temporal and the gene promoter resolution of the Maezawa study by defining 2 stages of H3K27me3 gain and 2 different gene promoter groups corresponding to these changes during adult spermatogenesis. In *Kdm2a* mutant SgD cells, the reduced level of H3K27me3 measured at cluster 3 gene promoters (Figure 5) suggests that the natural increase in H3K27me3 at the SgU to SgD transition (Supplemental Figure 10) does not occur in *Kdm2a* deficient SgD cells, making the cluster 3 genes susceptible for aberrant activation.

The impact of *Kdm2a* deficiency on H2AK119u1 levels awaits future investigations to understand the increased de-repression of cluster 2 genes. While we attempted ULI-NChIP for H2AK119ub1 in Ctrl and iKO SgD cells, the signal-to-noise ratio did not meet our internal quality control standards.

We present the new datasets in Supplemental Figure 10 and we have updated the main text accordingly.

Maezawa et al. 2018 H3K27me3 ChIP Hu et al. 2024 H2AK119ub Cut&Tag

Maezawa et al. 2018 H3K27me3 ChIP

Hu et al. 2024 H2AK119ub Cut&Tag

Rebuttal Figure 9. Heatmap showing H3K27me3 and H2AK119u1 of 5kb regions upstream/downstream relative to their TSSs. All genes were classified into 9 clusters shown in Figure 5A. From left to right: Cluster identity, Chromosomal location of the gene, Presence or absence of CpG island from the gene promoter, CpG density at gene TSS +/- 5kb, log₂RPKM values of H3K27me3 ChIP-seq for non-overlapping 50bp genomic windows located +/- 5kb from the gene TSS in THY1+ and cKIT+ spermatogonia isolated from juvenile mice, and pachytene spermatocytes isolated from adult mice (from Maezawa *et al.*, 2018), and log₂RPKM values of H2AK119u1 Cut&Tag for non-overlapping 50bp

genomic windows located +/- 5kb from TSS in undifferentiated and differentiated type A spermatogonia isolated from adult mice (from Hu *et al.*, 2024). H2AK119u1 Cut&Tag analysis is also presented in Supplemental Figure 10B,10D.

7. Given that some of the differentially expressed genes (DEGs) in Figure 4E encode important transcription factors (Pou5f1, Wt1, Zic2, Eomes, Lef1 and Twist1), I don't think that all DEGs are directly bound and regulated by KDM2A. It is better to clarify which DEGs are directly bound by KDM2A and analyse the changes in histone modification (e.g. H3K36me2, H3K4me3, H3K27me3) on these KDM2A-bound genes.

Thank you for your comment. In order to define target genes of KDM2A, we reanalyzed the recently published anti-KDM2A ChIP performed in p10 cKIT-positive germ cells by Feng *et al.* (2024) (6) which represents differentiating spermatogonia in juvenile animals. We have integrated this analysis into the heatmap of Main Figure 5A. We show that KDM2A is preferentially enriched at CGI-positive promoters (clusters 1-5) in agreement with its CXXC domain specificity (17). In addition, in Supplemental Figure 9C we show the anticorrelation between the KDM2A and H3K36me2 signal at CGI-positive promoters supporting the proposed demethylase function of KDM2A's Jumonji domain. In Main Figure 5 and its associated Supplemental Figure 9 we provide the reader a whole overview of gene regulation without focusing specifically at the DEGs.

Regarding the specific question on the DEGs, we plotted the normalized KDM2A-Input \log_2 RPKM signal of the TSS +/- 1kb and observed a bimodal distribution of gene promoters with a local minimum close to 0.5 which was used as a threshold to assign KDM2A enriched promoters > 0.5 \log_2 RPKM (Rebuttal Figure 10A). We defined 10437 (42.5%) gene promoters to be enriched for KDM2A (Rebuttal Figure 10B), mostly corresponding to genes with CpG islands or increase CpG density at their promoters (Main Figure 5A). 14091 (57.5%) genes did not show KDM2A enrichment (Rebuttal Figure 10B). Among the 778 upregulated genes in iKO vs Ctrl SgDs, we found 300 (38.5%) which exhibit high KDM2A signal (Rebuttal figure 10C). In particular promoters of important TFs, like *Wt1*, *Zic2*, *Eomes*, *Lef1* and *Twist1* were found to be enriched for KDM2A (Rebuttal Figure 10C). We then categorized the genes according to the transcriptional response in SgD iKO vs Ctrl and their promoter enrichment for KDM2A into 4 groups: a) non differentially expressed genes (DEGs) with KDM2A enrichment, b) non DEGs without KDM2A, c) Upregulated genes with KDM2A enrichment, d) Upregulated genes without KDM2A enrichment (Rebuttal Figure 10D-F). We assessed the change of H3K36me2 (Rebuttal Figure 10D), H3K4me3 (Rebuttal Figure 10E) and H3K27me3 (Rebuttal Figure 10F) at the promoters of the 4 genes groups as defined above in SgD iKO vs Ctrl samples. We found that the group of upregulated genes with KDM2A enrichment show on average the highest \log_2 FC of H3K36me2 and H3K4me3 and the lowest \log_2 FC of H3K27me3 when compared to the other gene groups.

Rebuttal Figure 10: A) Histogram showing the density of TSS +/- 1kb according to the KDM2A-Input log₂RPKM enrichment. B) Euler diagram showing the intersections between KDM2A enriched promoters and upregulated genes in Sgd iKO vs Ctrl. C) Scatterplot showing the log₂FC of gene expression in Sgd iKO vs Ctrl against the KDM2A-Input log₂RPKM at their promoters (TSS +/- 1kb). Red dashed line represents the cutoff to assign a promoter as KDM2A enriched. Cutoffs for DEGs were defined in the manuscript as log₂FC > 1 and FDR < 0.05. D-F) Boxplots representing the Sgd iKO vs Ctrl log₂FC of H3K36me2 (D), H3K4me3 (E) and H3K27me3 (F) of genes belonging to 4 groups based on their differential expression and KDM2A enrichment.

Minor comments:

1. Why does the *Wt1* gene, which encodes a Sertoli cell marker protein, appear in the set of genes that are up-regulated in KDM2A-ablated differentiated spermatogonia?

Indeed, *WT1* is a marker of Sertoli cells and required for gonad development (18). The *Wt1* gene promoter is marked by PRC2 / H3K27me3 in germ cells. PRC2 has been proposed to repress lineage specific genes in tissues in which they are not supposed to be expressed. Accordingly, we think that PRC2 marking of *Wt1* in germ cells keeps this gene repressed and KDM2A ablation impacts on this repression possibly by preventing PRC2 catalysis via H3K36me2. Importantly, other classical Sertoli cell specific gene markers are found to be very lowly expressed in our RNA-seq samples (Supplemental Figure 6G) arguing against sample contamination.

2. Line 944, the legend for Fig 4F should be corrected to "iKO ScLZ versus Ctrl ScLZ..."

Apologies for the mistyping. It is corrected accordingly.

Reviewer #6 (Remarks to the Author):

The authors have addressed Reviewer #3's comments comprehensively, particularly in providing mechanistic insights into the role of KDM2A in spermatogonial differentiation and meiotic progression. Below are my observations and recommendations:

We thank the reviewer for the positive evaluation of revised manuscript.

1. Mechanistic Insights:

The rebuttal effectively clarifies how KDM2A-specific effects, particularly its role in silencing PRC2 target genes, are pronounced in spermatogonial cells despite KDM2A's broader biological roles. The inclusion of ChIP-seq data on H3K36me2 and analyses of CGI-rich promoter dynamics strengthens the mechanistic narrative.

Thank you for commending our ChIP-seq data acquisition and analysis.

2. Histone Modification Dynamics:

The added ChIP-seq data on H3K36me2 addresses the reviewer's concerns, providing robust evidence for promoter-level changes in differentiating spermatogonia. The insights into changes in other histone marks (e.g., H3K4me3 and H3K27me3) also enrich the mechanistic understanding. Still, a deeper discussion of how these findings align with existing epigenetic paradigms in spermatogenesis would be beneficial.

Thank you for your comment. We have revised our discussion regarding the roles of Polycomb and DNA methylation during spermatogenesis. We now discuss our results in data in relationship to: PRC2-H3K27me3 in paragraph 3, PRC1- H2AK119u1 in paragraphs 4-6 and DNA methylation in paragraph 7.

3. Expression Pattern of KDM2A:

The addition of immunofluorescence data elucidating KDM2A expression during different spermatogenic stages is commendable. This visual evidence helps link KDM2A's presence with its functional role in differentiation and meiotic processes.

Thank you for appreciating the immunofluorescence experiments and analysis.

4. In vitro Validation via Stem Cell Culture:

While the authors' rationale for not pursuing in vitro experiments is reasonable, a more detailed explanation of limitations (e.g., challenges in synchronizing differentiation stages in culture) would add clarity.

In vitro experiments using spermatogonial stem cell (SSC) cultures are instrumental in advancing our understanding of spermatogenesis and developing fertility treatments. However, several limitations hinder their effectiveness:

Technical challenges: Setting up spermatogonial stem cell (SSC) cultures is technically demanding and requires specialized expertise. While protocols exist, optimizing and applying such complex culture media and conditions is challenging.

Loss of In Vivo Microenvironment: In vitro systems often fail to replicate the intricate microenvironment of the testis, which includes essential cell–cell interactions, tissue architecture, and physiological signals crucial for SSC maintenance and differentiation. The absence of supporting cells, such as Sertoli and Leydig cells, and vascular system disrupts the paracrine and autocrine signaling necessary for proper spermatogenesis.

Challenges in Synchronizing Differentiation Stages: Achieving synchronized progression through the various stages of spermatogenic differentiation in culture is difficult. This asynchrony can lead to heterogeneous cell populations, complicating the study of specific developmental stages and the identification of key regulatory factors.

Overall, the authors have addressed major concerns raised by Reviewer #3 thoughtfully and with substantial new data. I support to accept the revised manuscript.

Thank you for supporting the acceptance of our revised manuscript.

References

1. Hermann, B.P., Cheng, K., Singh, A., Roa-De La Cruz, L., Mutoji, K.N., Chen, I.C., Gildersleeve, H., Lehle, J.D., Mayo, M., Westernstroer, B. *et al.* (2018) The Mammalian Spermatogenesis Single-Cell Transcriptome, from Spermatogonial Stem Cells to Spermatids. *Cell reports*, **25**, 1650-1667 e1658.
2. Manova, K., Nocka, K., Besmer, P. and Bachvarova, R.F. (1990) Gonadal expression of c-kit encoded at the *W* locus of the mouse. *Development*, **110**, 1057-1069.
3. Schrans-Stassen, B.H., van de Kant, H.J., de Rooij, D.G. and van Pelt, A.M. (1999) Differential expression of c-kit in mouse undifferentiated and differentiating type A spermatogonia. *Endocrinology*, **140**, 5894-5900.
4. Yan, W., Kero, J., Huhtaniemi, I. and Toppari, J. (2000) Stem cell factor functions as a survival factor for mature Leydig cells and a growth factor for precursor Leydig cells after ethylene dimethane sulfonate treatment: implication of a role of the stem cell factor/c-Kit system in Leydig cell development. *Developmental biology*, **227**, 169-182.
5. Turberfield, A.H., Kondo, T., Nakayama, M., Koseki, Y., King, H.W., Koseki, H. and Klose, R.J. (2019) KDM2 proteins constrain transcription from CpG island gene promoters independently of their histone demethylase activity. *Nucleic acids research*, **47**, 9005-9023.
6. Feng, S., Gui, Y., Yin, S., Xiong, X., Liu, K., Li, J., Dong, J., Ma, X., Zhou, S. and Zhang, B. (2024) Histone demethylase KDM2A recruits HCFC1 and E2F1 to orchestrate male germ cell meiotic entry and progression. *The EMBO journal*, 1-31.
7. Xiong, X., Huang, X., Zhu, Y., Hai, Z., Fei, X., Pan, B., Yang, Q., Xiong, Y., Fu, W., Lan, D. *et al.* (2023) Testis-specific knockout of *Kdm2a* reveals

- nonessential roles in male fertility but partially compromises spermatogenesis. *Theriogenology*, **209**, 9-20.
8. Kanatsu-Shinohara, M., Tanaka, T., Ogonuki, N., Ogura, A., Morimoto, H., Cheng, P.F., Eisenman, R.N., Trumpp, A. and Shinohara, T. (2016) Myc/Mycn-mediated glycolysis enhances mouse spermatogonial stem cell self-renewal. *Genes & development*, **30**, 2637-2648.
 9. Murta, D., Batista, M., Silva, E., Trindade, A., Henrique, D., Duarte, A. and Lopes-da-Costa, L. (2013) Dynamics of Notch pathway expression during mouse testis post-natal development and along the spermatogenic cycle. *PLoS one*, **8**, e72767.
 10. Pesce, M., Wang, X., Wolgemuth, D.J. and Scholer, H. (1998) Differential expression of the Oct-4 transcription factor during mouse germ cell differentiation. *Mechanisms of development*, **71**, 89-98.
 11. Sharma, M., Srivastava, A., Fairfield, H.E., Bergstrom, D., Flynn, W.F. and Braun, R.E. (2019) Identification of EOMES-expressing spermatogonial stem cells and their regulation by PLZF. *eLife*, **8**.
 12. Huang, Z., Rivas, B. and Agoulnik, A.I. (2013) NOTCH1 gain of function in germ cells causes failure of spermatogenesis in male mice. *PLoS one*, **8**, e71213.
 13. Zheng, Y., Phillips, L.J., Hartman, R., An, J. and Dann, C.T. (2016) Ectopic POU5F1 in the male germ lineage disrupts differentiation and spermatogenesis in mice. *Reproduction*, **152**, 363-377.
 14. Nyfeler, Y., Kirch, R.D., Mantei, N., Leone, D.P., Radtke, F., Suter, U. and Taylor, V. (2005) Jagged1 signals in the postnatal subventricular zone are required for neural stem cell self-renewal. *The EMBO journal*, **24**, 3504-3515.
 15. Hu, M., Yeh, Y.H., Maezawa, S., Nakagawa, T., Yoshida, S. and Namekawa, S.H. (2024) PRC1 directs PRC2-H3K27me3 deposition to shield adult spermatogonial stem cells from differentiation. *Nucleic acids research*, **52**, 2306-2322.
 16. Maezawa, S., Hasegawa, K., Yukawa, M., Kubo, N., Sakashita, A., Alavattam, K.G., Sin, H.S., Kartashov, A.V., Sasaki, H., Barski, A. *et al.* (2018) Polycomb protein SCML2 facilitates H3K27me3 to establish bivalent domains in the male germline. *Proceedings of the National Academy of Sciences of the United States of America*, **115**, 4957-4962.
 17. Blackledge, N.P., Zhou, J.C., Tolstorukov, M.Y., Farcas, A.M., Park, P.J. and Klose, R.J. (2010) CpG islands recruit a histone H3 lysine 36 demethylase. *Molecular cell*, **38**, 179-190.
 18. Gao, F., Maiti, S., Alam, N., Zhang, Z., Deng, J.M., Behringer, R.R., Lecureuil, C., Guillou, F. and Huff, V. (2006) The Wilms tumor gene, *Wt1*, is required for Sox9 expression and maintenance of tubular architecture in the developing testis. *Proceedings of the National Academy of Sciences of the United States of America*, **103**, 11987-11992.

REVIEWERS' COMMENTS

We sincerely thank all the reviewers for their time and effort in evaluating our revised manuscript.

Reviewer #4 (Remarks to the Author):

My questions have all been adequately addressed! It is now suitable for acceptance. We are grateful to the reviewer for their positive assessment and are pleased to know that all concerns have been adequately addressed.

Reviewer #5 (Remarks to the Author):

The authors have comprehensively addressed the comments and discussed the differences between the current study and other previous publications. Despite the fact that the originality of the work has been compromised to some extent, it is nevertheless deserving of publication within this field.

We thank the reviewer for their thoughtful feedback and constructive evaluation. We acknowledge the comment regarding the originality of the work and respectfully recognize the reviewer's perspective.

At present, the literature presents conflicting findings regarding the role of KDM2A in murine spermatogenesis. Notably, **Xiong et al. (2023)** reported that “*Surprisingly, Kdm2a-deficient adult males were completely fertile and comparable with their control ($Kdm2a^{flox/flox}$) counterparts,*” whereas **Feng et al. (2024)** concluded that “*Conditional deletion of Kdm2a in mouse pre-meiotic germ cells results in complete male sterility, with spermatogenesis ultimately arrested at the zygotene stage of meiosis.*”

We believe that our study contributes meaningful clarification to this controversy by offering additional experimental evidence and providing new mechanistic insights. Specifically, we argue that “*Using timed conditional deletion approaches, we demonstrate that Kdm2a deficiency causes testicular atrophy and male infertility. While spermatogonial stem cells were unaffected, proliferating and differentiating spermatogonia suffered from delayed cell cycle progression and apoptosis. Our data points to novel roles for the catalytic activity of Kdm2a in controlling Polycomb-mediated gene repression during spermatogonial differentiation.*”

We hope this contribution provides valuable context to the ongoing debate and strengthens the understanding of KDM2A's role in male germ cell development.